# Longest sediment flows yet measured show how major rivers connect efficiently to deep sea

Peter J. Talling [1✉], Megan L. Baker [2], Ed L. Pope [2], Sean C. Ruffell [3], Ricardo Silva Jacinto[4], Maarten S. Heijnen[5,6], Sophie Hage [7,8], Stephen M. Simmons [9], Martin Hasenhündl [10], Catharina J. Heerema[3], Claire McGhee[11], Ronan Apprioual[4], Anthony Ferrant[4], Matthieu J. B. Cartigny [2], Daniel R. Parsons [9], Michael A. Clare [5], Raphael M. Tshimanga[12], Mark A. Trigg[13], Costa A. Cula[14], Rui Faria[14], Arnaud Gaillot[4], Gode Bola[12], Dec Wallance[15], Allan Griffiths[16], Robert Nunny [17], Morelia Urlaub [18], Christine Peirce [3], Richard Burnett[19], Jeffrey Neasham[19] & Robert J. Hilton[20]

Here we show how major rivers can efficiently connect to the deep-sea, by analysing the longest runout sediment flows (of any type) yet measured in action on Earth. These seafloor turbidity currents originated from the Congo River-mouth, with one flow travelling >1,130 km whilst accelerating from 5.2 to 8.0 m/s. In one year, these turbidity currents eroded 1,338-2,675 [>535–1,070] Mt of sediment from one submarine canyon, equivalent to 19–37 [>7–15] % of annual suspended sediment flux from present-day rivers. It was known earthquakes trigger canyon-flushing flows. We show river-floods also generate canyon-flushing flows, primed by rapid sediment-accumulation at the river-mouth, and sometimes triggered by spring tides weeks to months post-flood. It is demonstrated that strongly erosional turbidity currents self-accelerate, thereby travelling much further, validating a long-proposed theory. These observations explain highly-efficient organic carbon transfer, and have important implications for hazards to seabed cables, or deep-sea impacts of terrestrial climate change.

[1] Departments of Geography and Earth Science, Durham University, South Road, Durham DH1 3LE, UK. [2] Department of Geography, Durham University, South Road, Durham DH1 3LE, UK. [3] Department of Earth Sciences, Durham University, South Road, Durham DH1 3LE, UK. [4] Marine Geosciences Unit, IFREMER Centre de Brest, Plouzané, France. [5] National Oceanography Centre Southampton, SO14 3ZH Southampton, UK. [6] School of Ocean and Earth Sciences, University of Southampton, Southampton SO14 3ZH, UK. [7] University of Brest, CNRS, IFREMER, Geo-Ocean, 29280 Plouzané, France. [8] Department of Geosciences, University of Calgary, Calgary, AB T2N 1N4, Canada. [9] Energy and Environment Institute, University of Hull, Hull HU6 7RX, UK. [10] Institute of Hydraulic Engineering and Water Resources Management, TU Wien, 1040 Vienna, Austria. [11] School of Civil Engineering and Geosciences, Newcastle University, Newcastle upon Tyne, UK. [12] Congo Basin Water Resources Research Center (CRREBaC) and Department of Natural Resources Management, University of Kinshasa (UNIKIN), Kinshasa, Democratic Republic of the Congo. [13] School of Civil Engineering, University of Leeds, Leeds LS3 9JT, UK. [14] Angola Cables SA, Cellwave Building 2nd Floor Via AL5, Zona XR6B, Talatona-Luanda, Angola. [15] Subsea Centre of Excellence Technology, BT, London, UK. [16] O&M Submarine Engineering, Vodaphone Group, Leeds, UK. [17] Ambios, 1 Hexton Road, Glastonbury, Somerset BA6 8HL, UK. [18] GEOMAR Helmholtz Centre for Ocean Research, Wischhofstraße 1-3, 24148 Kiel, Germany. [19] School of Engineering, Newcastle University, Newcastle upon Tyne NE1 7RU, UK. [20] Department of Earth Sciences, South Parks Road, Oxford OX1 3AN, UK. ✉email: Peter.J.Talling@durham.ac.uk

Flows of sediment that move along the seabed (called turbidity currents) form the largest sediment accumulations, deepest canyons and longest channel systems on Earth[1-3]. The scale of individual turbidity currents can also be exceptionally large (Table 1). For example, an earthquake-triggered turbidity current that occurred in 1929 in the NW Atlantic carried over 200 km$^3$ of sediment, and ran out for >800 km, at speeds of up to 19 m/s[4,5]. This single turbidity current carried over 14 times the modern-day annual suspended sediment flux from all of the world's rivers[6] (Table 1), and its volume exceeded the largest documented terrestrial landslide in the last ~350,000 years[7]. It was previously thought that directly measuring powerful turbidity currents that reached the deep-sea was impractical[8]. However, here we describe direct monitoring of deep-sea turbidity currents in the Congo Canyon offshore West Africa[9], whose timing was captured by an array of seabed moorings and seabed telecommunication cable breaks (Figs. 1 and 2). On January 14–16th 2020, one of these flows travelled for over 1130 km from the mouth of the Congo River, measured along the sinuous axis of the submarine Congo Canyon and Channel (Fig. 1). This is the longest runout sediment-driven flow yet measured in action, with a runout distance exceeding that of the 1929 NE Atlantic turbidity current[4], and longest known terrestrial debris flow[10], snow avalanche[11] or volcanic pyroclastic flow[12] (Table 1).

The scale of turbidity currents ensures that the sediment-mass carried by these flows rivals that of any other process on Earth[1,13], including rivers[6] or glaciers[6], or settling from the surface ocean[14] (Table 1). Turbidity currents are thus important for a wide variety of reasons. For example, turbidity currents play a globally significant role in terrestrial organic carbon burial[15] that affects atmospheric $CO_2$ levels over geological time scales, and other global geochemical cycles. It was once thought that terrestrial organic carbon was primarily oxidised on continental shelves[16]. More recent studies[15,17] proposed that transfer and burial of terrestrial organic carbon in the deep-sea by turbidity currents might be highly efficient, based on similar organic carbon abundance, composition and age in sediment samples from rivermouths and upper-canyons[17] or deep-sea channels[15]. However, these studies[15,17] did not document how such efficient sediment and organic carbon transfer actually occurred. Here, we use direct observations to explain why transfer of sediment and associated organic carbon from rivers to the deep-sea can be so efficient. Organic carbon also forms the basis for most seafloor food webs, and rapid and sustained deposition of organic-rich sediment by turbidity currents can create distinct ecosystems, such as at the end of the Congo system[18,19]. These sometimes very powerful flows can also scour life from floors of submarine canyons[20], and this study therefore also illustrates how turbidity currents affect deep-sea life in disparate ways.

Turbidity currents are also important geohazards[21]. In particular, they break seabed telecommunications cable networks that now carry over 99% of intercontinental data traffic[22], which underpin the global internet and many other aspects of our daily lives worldwide[23-25]. The January 2020 flow described here broke both telecommunication cables (Figs. 1 and 2) connecting to West Africa, causing the internet to slow significantly from Nigeria to South Africa[9], and these cables were broken again by turbidity currents in March 2020, April 2021 and January 2022 (Supplementary Table 1), including during coronavirus (CoV-19) related lockdown when internet bandwidth was particularly critical. Understanding why these cables broke is important, especially as they had not broken in the preceding 18 years (Supplementary Table 1). It has been proposed that turbidity current deposits (turbidites) can provide long-term records of other major hazards, including earthquakes, typhoons or river floods[26-28]. Such records are potentially valuable, as they extend further back in time than most records on land. This study provides detailed information on how long-runout turbidity currents are related to river floods, and how floods are recorded in the deep-sea.

**Table 1 Comparison of Congo Canyon turbidity currents and other types of sediment flow or global fluxes.**

| Sediment volume/mass and runout distance of individual events | Sediment volume transported (km$^3$) | Runout distance (km) |
|---|---|---|
| Congo Canyon Turbidity Currents in 2019–20 (this study) (*sediment volume and mass eroded from seabed; based on time-lapse seabed surveys in Sept–Oct. 2019 to Oct. 2020*)(*1.07 km$^3$ eroded in survey length that is 40% of total length*) | ~2.675 km$^{3a}$ (1338–2675 Mt)$^b$ | >1130 km |
| Grand Banks turbidity current in 1929, N.W. Atlantic[4]. | >200 km$^3$ (100,000– 200,000 Mt)$^b$ | >800 km |
| Mt. St. Helens landslide in 1980: largest historical landslide[7] | 2.8 km$^3$ | 22.5 km |
| Largest snow avalanches[11] | 0.01 km$^3$ | <3–5 km |
| AD184 Taupo pyroclastic flows—largest volcanic pyroclastic flows in last 2000 years[12] | 30 km$^3$ | <90 km |
| Longest terrestrial lahar or debris flows in last century[10] | - | <90 km |
| Sediment flux by turbidity currents to deep sea after $M_w$ 9.1 Tōhoku earthquake[45] | 0.2 km$^3$ | 200–500 km |
| Sediment flux by turbidity currents to deep sea after $M_w$ 7.8 Kaikōura earthquake[20] | 0.94 km$^3$ | >700 km |
| *Global or local annual sediment fluxes* | *Sediment mass* | |
| Congo River—suspended sediment load[6,55] | ~29–43 Mt/yr | – |
| Congo River—bedload[54] | Up to 130 Mt/yr | – |
| Rivers (suspended sediment load): modern-day (2010)[6] | ~7200 Mt/yr | – |
| Rivers (suspended sediment load): pre-Anthropocene[68] | ~15–18,000 Mt/yr | – |
| Rivers (bedload—but very poorly known): modern day[6,68] | ~720−300 Mt/yr | – |
| Rivers (dissolved load) pre-Anthropocene & modern day[6,68] | ~3600–3800 Mt/yr | – |
| Sediment settling from surface ocean[14] | ~54,600 Mt/yr | – |
| Sediment settling from surface ocean that reaches the seabed[14] | ~2960 Mt/yr | – |
| Aeolian dust transport from land to oceans[6] | ~1500 Mt/yr | - |
| Glacial transport (icebergs and meltwater): modern day[6] | ~ 4000 Mt/yr | - |

$^a$Surveys that recorded 1.07 km$^3$ [>0.40 km$^3$] of erosion only covered 40% of the total canyon-channel length, suggesting that 2.675 km$^3$ [>1.00 km$^3$] of seabed erosion occurred along its entire length (see 'Methods' section).
$^b$This is based on global average of porosity of ~60–80% in upper 50 m of seabed sediment[70], a grain density of ~2500 kg/m$^3$, and thus a (dry) seafloor sediment density of ~500 to 1000 kg/m$^3$ (see 'Methods' section).

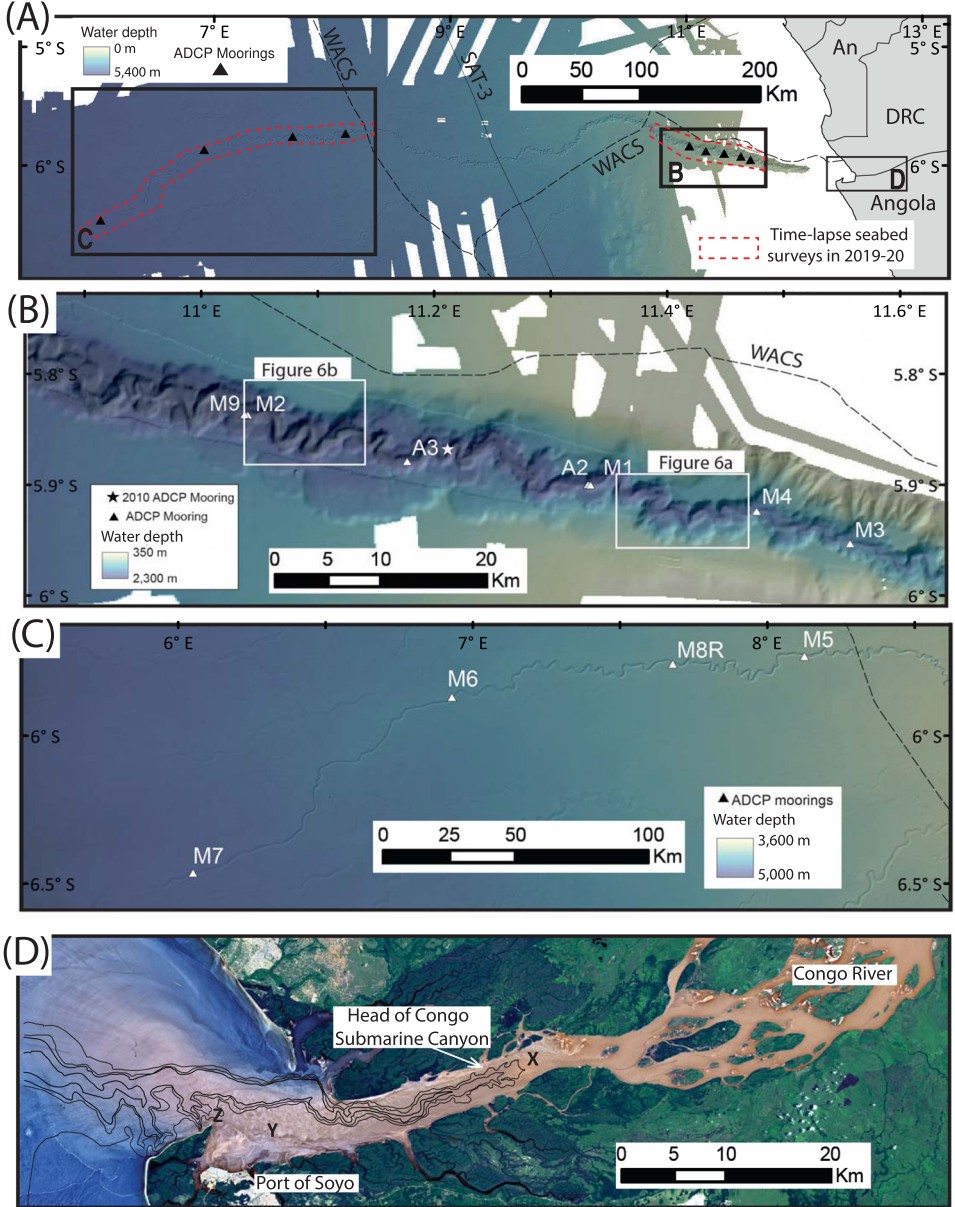

**Fig. 1 Location map of oceanographic moorings and telecommunication cables that recorded turbidity currents in 2019–21 in the Congo Canyon and Channel, offshore from the mouth of the Congo River in West Africa. A** Map of the entire array with mooring (e.g., M1 or A2) and cable (e.g., WACS) names. Red dotted lines indicate areas where time-lapse bathymetric surveys were collected in September–October 2019 and October 2020. **B**, **C** Detailed map of the upper submarine canyon, and deep-water submarine channel, with locations in **A**. **D** The head of the Congo Submarine Canyon lies within the estuary forming the mouth of the Congo River, with the river producing a surface plume of sediment that extends offshore. Landsat 8 image on 02-03-2015 with superimposed bathymetric contour at 20, 100, 200, and 400 m from UK Admiralty Chart 658. The main submarine canyon head (x), a shallow-water plateau off Soyo (y), and tributary canyon heads (z) are indicated.

Despite their importance, there are remarkably few direct measurements from turbidity currents, ensuring they are poorly understood[1]. This is a stark contrast to far more numerous and widespread direct measurements from of other major sediment transport processes[6,14,29]. Recent pioneering work has shown how short runout (<~50 km) turbidity currents can be monitored in shallow water, typically using moorings with sensors, such as acoustic Doppler current profilers (ADCPs) that measure profiles of flow velocity and sediment backscatter[30–35]. However, detailed monitoring is still only available for turbidity currents at <10 sites worldwide, all in water depths of <2 km[30–35], and for flows that ended within and infilled canyons. There were no detailed direct measurements for far more powerful and erosive 'canyon-

flushing' turbidity currents, which carry sediment beyond the canyon's end, and dominate longer-term sediment transfer. This situation ensured that fundamental questions remain. For example, previous studies showed that major earthquakes can sometimes trigger canyon-flushing turbidity currents that carry very large amounts of sediment[4,20]. However, it was not clear whether river floods also generate such large canyon-flushing events[27] (Supplementary Discussion). It was also theorised that turbidity currents behave in a very different way to rivers; as turbidity currents that erode the seabed could become denser and faster, and erode yet more sediment and become even denser, causing turbidity currents to self-accelerate or 'ignite'[36]. However, sustained ignition was yet to be documented clearly in submarine

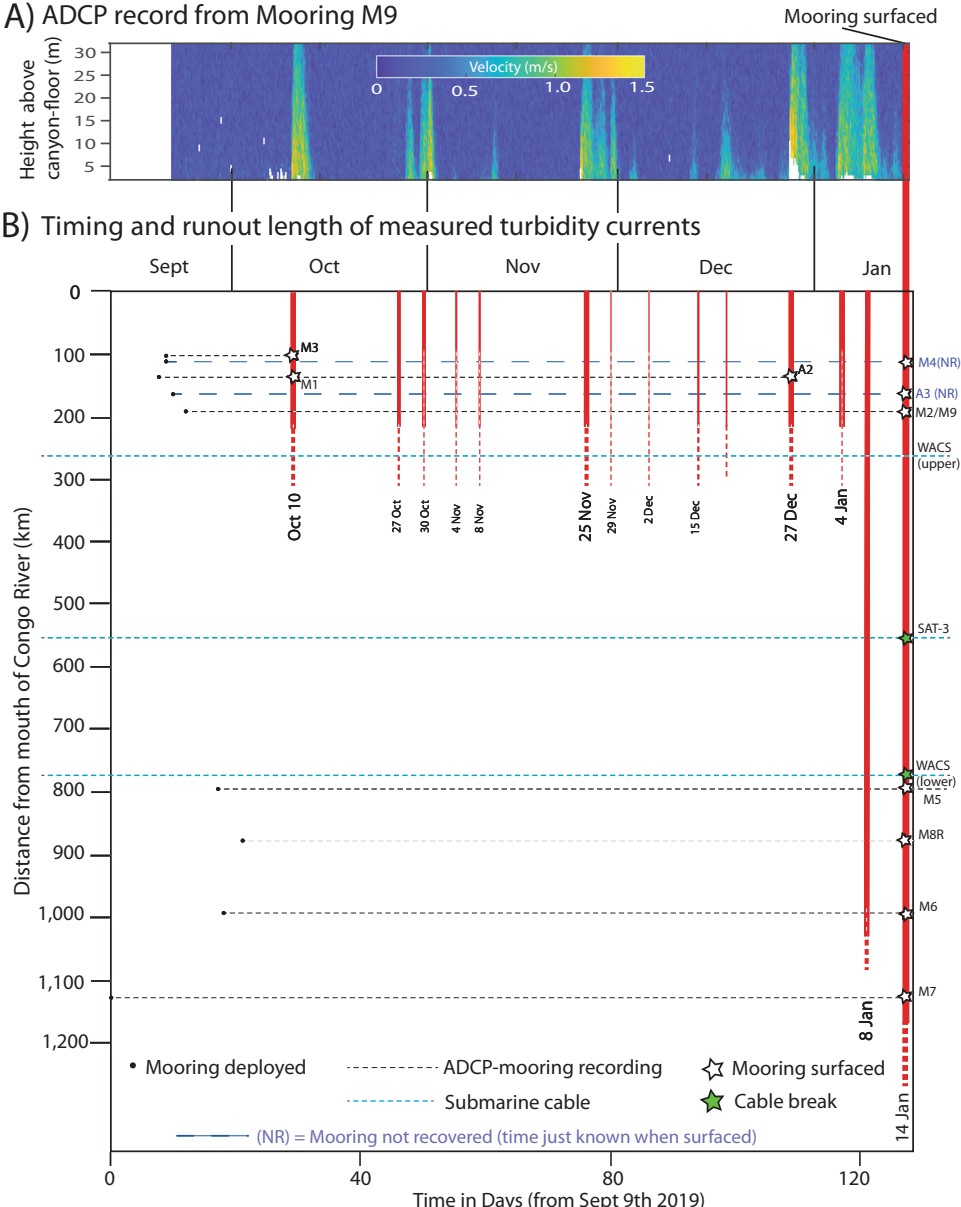

**Fig. 2 Timing and runout distance of turbidity currents measured from September 2019 to January 2020 along the Congo Canyon and Channel system. A** ADCP time series of velocities measured at mooring M9, showing occurrence of turbidity currents. **B** Plot of event timing against distance from Congo River mouth, as measured along the sinuous canyon-channel axis. Red vertical lines denote flow events (dotted where termination uncertain), and indicate their runout distances, with the most powerful January 14–16[th] event in bold. Dotted horizontal lines denote a mooring site or submarine cable. The times of mooring deployment are shown, together with when moorings or cables broke due to turbidity currents. Two moorings (M4 and A3) were not recovered ('NR'); flow timings at these two sites are derived from when mooring reached the ocean surface, and an assumed rise rate of 150 m/min (as seen during earlier work).

flows[35], and it was unclear what factors determined whether it occurred.

Here we show how turbidity currents connect major rivers to the deep-sea, by presenting the first detailed measurements from turbidity current within the deep (2–5 km) ocean, which combines information from cable breaks with that from 9 ADCP-moorings along a 900 km length of the Congo Canyon and Channel[9] (Figs. 1 and 2). First, we seek to understand how unusually powerful and long-runout turbidity currents are initiated that flush submarine canyons, and what controls their timing. Canyon-flushing flows are found to be associated with major river floods, but finally triggered weeks to months after the flood peak, typically at spring tides. Second, we seek to understand how

turbidity currents behave, and why some flows accelerate and runout much further. It is shown that oceanic turbidity currents can accelerate ('ignite'), in sometimes for a thousand kilometres, as they erode prodigious volumes of seabed sediment. There is a threshold initial front speed (>4–5 m/s) for long-runout flows, but this threshold speed is weakly controlled by sediment grain-size, contrary to past theory. These findings underpin a generalised model for how turbidity currents transfer globally significant sediment volumes from major rivers to the deep-sea. Finally, the wider implications of this study are outlined for efficiency of organic carbon transfer to the deep-sea[15–17], predicting hazards to seabed telecommunication cables[9,22–25], and how future climate or land-use changes may impact the deep-sea.

## Results

**Study area.** The head of the Congo Submarine Canyon lies within the estuary of the Congo River (Fig. 1D), which has the second largest water discharge and fifth largest particulate organic carbon export of any river[6]. The canyon incises deeply into the continental shelf and slope, before transitioning in a less-deeply incised conduit termed a submarine channel[37–39] (Fig. 1A–C). The channel terminates at a water depth of ~4800 m, beyond which there is an area of sediment deposition termed a lobe[39,40]. Previous deposit-based studies suggest long-term sediment transfer through the canyon and channel is efficient, with ~30% of the total sediment mass located in lobe deposits[40,41]. Exceptionally rapid deposition of organic carbon-rich (3–4% TOC) sediment of mainly terrestrial origin (70–90%) leads to efficient organic carbon burial on the lobe[40,41], with methane-rich fluids due to diagenesis of this organic matter leading to unusual chemosynthesis-based ecosystems[18,19].

Past work along the Congo Canyon produced some of the first measurements from turbidity currents, albeit with less-detailed sensors[38,42], or at just one site in the upper canyon[31–33]. Initially, current metres recorded velocities at a single height above the seabed, at three sites along the canyon-channel[38,42]. These measurements were averaged over an hour, and flow velocities reached up to ~1 m/s, before moorings broke[38,42]. This work documented transit speeds between moorings that decreased from 3.5 to 0.7 m/s[42]. Subsequently, moored ADCPs were used to record more detailed (every ~30 s) velocity profiles through flows in the upper canyon in 2010–13[31–33]. However, no previous study had deployed ADCP-moorings at multiple sites to the end of a deep-sea canyon-channel, as occurred during this 2019–2020 project[9] (Figs. 1 and 2). Eleven ADCP-moorings were deployed at depths of 1560 to 4730 m (Fig. 1), with each mooring containing one or more ADCPs, located 30 to 150 m above the seabed[9] (Supplementary Fig. 1).

**Initial causes of powerful and very long-run-out turbidity currents.** Twelve flows restricted to the upper canyon were recorded by ADCP-moorings between September 2019 and early January 2020 (Fig. 2), causing three moorings to break. A much longer and more powerful flow then occurred on 14–16th January 2020, breaking the eight remaining moorings and two seabed telecommunication cables (Fig. 2; Supplementary Tables 1 and 2). Data from 9 of 11 ADCP-moorings were recovered successfully, despite considerable challenges as surfaced moorings drifted across the sea-surface, amid CoV-19 related lockdowns. Further cable breaks due to turbidity currents occurred on March 9th 2020, April 28–29th 2021, and January 28th 2022 (Supplementary Table 1). No cable breaks had occurred in the preceding ~18 years, despite one or more cables being present in the canyon during this period (Supplementary Table 1), suggesting cable-breaking flows in 2020–2022 were unusually powerful.

None of the turbidity currents recorded by the ADCPs or cable breaks coincided with earthquakes, and there is no clear relation to offshore wave heights (Supplementary Material). However, these cable-breaking flows are associated with the largest floods of the Congo River since the early 1960s, and they occurred after an 18 year period without cable-breaks or comparable floods. A 1-in-50 year flood occurred with a peak discharge of ~70,883 m³/s at Kinshasa on December 21st 2019 (Fig. 3), with the flood peak most likely arriving ~2–4 days later at the river-mouth estuary[43]. The first cable-breaking flow occurred on January 14–16th, 3 weeks after the flood peak on December 21st, albeit when river discharge was still relatively high (Fig. 3B). The arrival times of this January 14–16th turbidity current were captured by eight ADCP-moorings just before they broke. The second cable-breaking flow on March 9th 2020, occurred 10 weeks after the flood peak while river discharge was lower (Fig. 3B). A second major (1-in-20 year) flood occurred the following year, with a peak discharge of 67,210 m³/s in Kinshasa on December 13th 2020[43]. This was followed by a third cable-breaking flow on April 28–29th 2021, some 4.5 months after the December 2020 flood. A fourth flow broke cables on 28th January 2022, ~6 weeks after a modest (54,651 m³/s) annual peak in river discharge (Fig. 3). There were significant delays between the flood peaks and the cable-breaking flows, and three of the four cable-breaking flows coinciding with subsequent spring tides (Fig. 4). It appears that floods supplied large amounts of sediment that primed the river mouth to produce powerful and long-runout flows (Fig. 3), which were triggered finally 3 weeks to 4.5 months after flood peaks, sometimes at spring tides (Fig. 4).

**Flow behaviour.** Changes in turbidity current transit (front) speeds, and flow behaviour, are documented by arrival times at ADCP-moorings and cable breaks. These data show that the front of the January 14–16th turbidity current progressively accelerated as it ran out for over 1130 km (Fig. 5A). The flow-front initially moved at 5.0–5.2 m/s for its first 500 km, before reaching a velocity of 8.2 m/s over 1000 km from source, albeit with a decrease in front speed between ~880 and 1000 km (Fig. 5A).

ADCP-moorings recorded a further 13 flows between September 2019 and January 2020 (Figs. 2 and 5). Twelve of these flows terminated in the upper canyon, and these events had front velocities of <4 m/s (Fig. 5A). One flow on January 5–15th travelled for >800 km, initially with a front speed of 4.4 m/s, but this flow decelerated to speeds of <1 m/s in deep-water, and terminated before the final mooring (Fig. 5A). Cable breaks on 28–29th April 2021 recorded a long-runout flow travelling at 4.0 m/s, although no ADCP-moorings remained to capture this event in detail (Fig. 2). Thus, a broad pattern emerges; flows with initial front speed exceeding 4 m/s ran out for long distances (>1000 km), and accelerated if their initial front speed was ≥5.0 m/s. In contrast, initially slower (<4 m/s) moving flows decelerated and ran out for 200–800 km (Figs. 2 and 3).

**Associated seafloor erosion.** Time-lapse surveys in September–October 2019 and October 2020 show that 1.07 km³ [>0.40 km³] was eroded from resurveyed reaches (Fig. 1) of the upper canyon and deep-water channel. We report eroded volumes in the form of $X$ [>$Y$] where $X$ is a most probable value and $Y$ is a conservative minimum estimate (see 'Methods' section). The resurveyed reaches comprise only 40% (477 of 1179 km) of the total canyon-channel length (Fig. 1A), so the total amount of eroded seabed sediment may be 2.68 km³ [>1.00 km³] (Supplementary Table 4). This is an exceptionally large sediment volume with a mass of ~1338–2675 Mt [>500–1000 Mt]. For comparison, currently the global annual suspended sediment flux from rivers is ~7200 Mt (Table 1)[6]. The unusually powerful turbidity currents in January and March 2020 presumably caused this erosion. The amount of sediment eroded along the flow pathway probably greatly exceeds that initially within these flows, as the eroded mass is 31-to-92 times the average annual suspended sediment supply from the Congo River (29–43 Mt/yr)[6,44] (Table 1; Supplementary Discussion).

## Discussion

It was previously known that major earthquakes could generate powerful and long-runout turbidity currents that transfer very large volumes of sediment to the deep-sea[4,5,20]. However, it was uncertain whether river floods could also generated turbidity currents that flushed large amounts of sediment through canyons

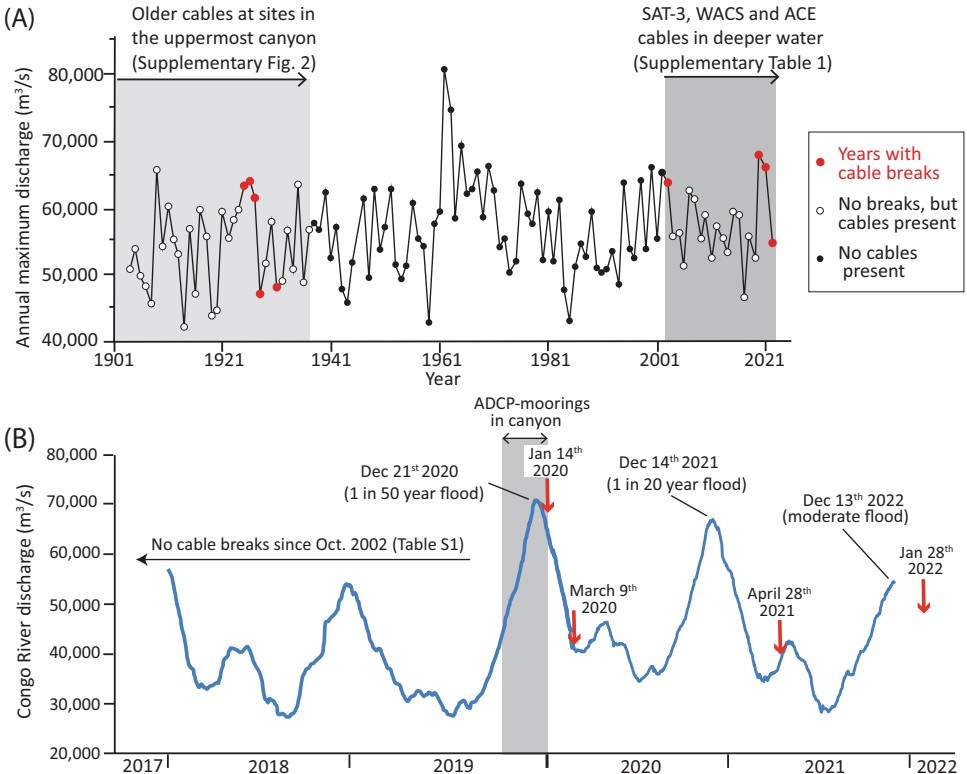

**Fig. 3 Turbidity currents that break cables are associated with major floods along the Congo River. A** Time series of annual maximum discharge of the Congo River, measured at Kinshasa. A river discharge of 70,883 m³/s occurred on December 21st 2019 that represents a 1-in-50 year flood, while a discharge of 67,210 m³/s occurred on December 12th 2020 that represent a 1-in-20 year flood. The SAT-3 submarine telecommunication cable had been in operation since 2001, and the last cable break before January 14th 2020 due to a turbidity current was in October 2001. **B** Daily Congo River discharge at Kinshasa from January 2018 to March 2020, showing timing of cable-breaking flows on 14th January and 9th March 2020, 28th April 2021, and January 28th 2022. There are delays of 3 weeks to 4.5 months between flood peaks and cable-breaking turbidity currents. The grey box shows when ADCP-moorings were deployed in the Congo Canyon during this project (Fig. 2).

to the deep-sea (Supplementary Discussion); and if so, how this occurred.

Here we document directly that major river floods generate powerful and long-runout large turbidity currents that flush very large amounts of sediment through submarine canyons. Indeed, the turbidity currents that flushed the Congo Canyon-Channel in January and March 2020 eroded ~2.68 km³ [>1.00 km³] or 1338–2675 Mt [>500–1000 Mt] of seabed sediment (Table 1). This mass is equivalent to 19–37% of the present-day annual suspended sediment flux (~7200 Mt) from all rivers[6] (Table 1) and it was carried down a single submarine canyon-channel, probably by just two turbidity currents (Fig. 2). The 1929 event in the NW Atlantic[4] involved a much larger sediment volume (>200 km³), but the amount of sediment carried by flood-related events in Congo Canyon rivals or exceeds other turbidity currents due to earthquakes, such as those offshore New Zealand in 2016 (1 km³ [>0.4 km³]; $M_w$ 6.8 Kaikōura earthquake[20]) or Japan in 2011 (~0.2 km;³ $M_w$ 9.1 Tōhoku earthquake[45]). Turbidity currents in 2020 and 2021 that flushed the Congo Canyon-Channel were linked to two river floods with recurrence intervals of 20 and 50 years[43]. This flood recurrence interval is significantly shorter than recurrence intervals of major earthquakes (100–300 years) that were previously proposed to trigger canyon-flushing events elsewhere[20,45–47].

Turbidity currents that flushed the Congo Canyon were associated with river floods, and in most cases spring tides (Figs. 3 and 4). Recent studies have shown how elevated river discharge and tides can combine to generate much shorter runout (1–50 km) turbidity currents offshore from smaller river

mouths[30,48–50], and how the threshold suspended sediment concentration of rivers needed for offshore flows is much lower than once thought[48] (Supplementary Discussion). However, this study shows that floods and tides can also generate far larger turbidity currents offshore from one of the world's largest rivers, and in an estuarine setting. This suggests that floods and tides may trigger turbidity currents in an even wider range of settings than previously thought, which then transfer globally significant sediment volumes.

Delays of several weeks to months occur between river floods and turbidity currents that flush the Congo Canyon (Fig. 3). Previous work documented significant delays between river floods and associated turbidity currents, but only for hours[49] to days[24,25], not weeks to months. This suggests that river-mouths can store flood-sediment for up to several months, and maybe years, and thus act as an efficient 'capacitor', before eventually releasing sediment in one or more long-runout turbidity currents.

The January 2022 event occurred after a moderate peak in annual Congo River discharge, and not at a spring tide (Figs. 3 and 4). Long-runout turbidity currents can therefore also be caused by smaller floods, and this is also shown by cable-breaks off Taiwan in 2015 after Typhoon Soudelor (Supplementary Fig. 6). It is possible that preceding much larger river floods supplied sediment that contributed to generating long-runout turbidity currents in later years. Older cable breaks (1883 to 1937) in the Congo Canyon[9] also indicate clusters of cable breaks may occur for several years after major floods (Supplementary Fig. 2).

Past work on how floods cause turbidity currents has often focussed on a model in which the floodwater has enough

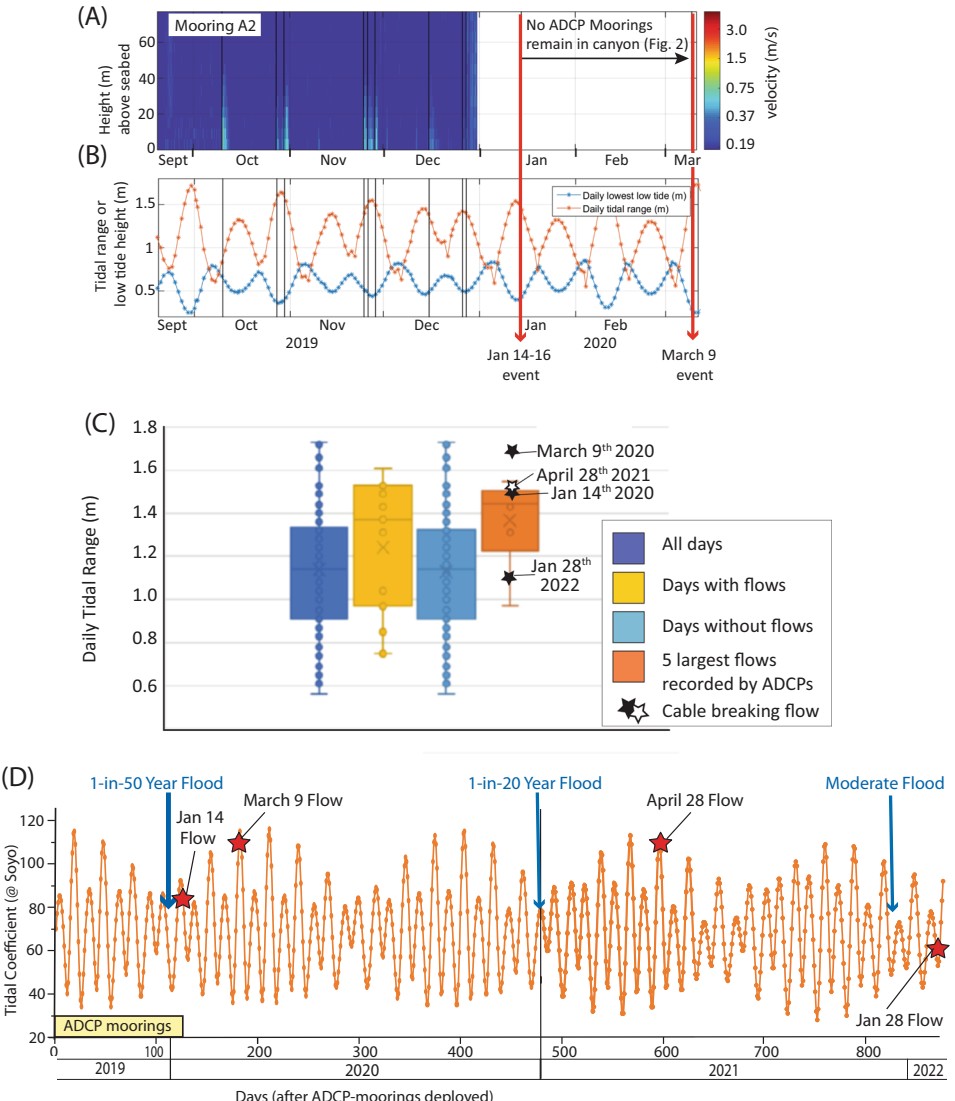

**Fig. 4 Cable-breaking turbidity currents coincide with spring tides. A** Time series of velocity profiles recorded by an ADCP at mooring site A2 in 2019–20 (Fig. 1), with warmer colours indicating turbidity currents. Superimpose are the arrival timings of long-runout, cable-breaking turbidity currents on January 14th and March 9th 2020 (red lines), and slower moving flows restricted to the upper canyon (thin black lines). **B** Time series of daily maximum tidal range, and daily lowest low tide, at Soyo in the estuary at the mouth of the Congo River. **C** Box and whisker plots showing median, first and second quartiles of daily tidal range values for (i) all days in which ADCP-moorings were in the Congo Canyon in 2019–2020, (ii) days on which turbidity currents occurred at the ADCP moorings, (iii) days on which no turbidity currents occurred at ADCP moorings, and (iv) days on which the five fastest non-cable-breaking flows occurred at ADCP moorings. Each box and whisker plot shows the median tidal range (x), tidal ranges on given days (o), and the 95% percentile of the distribution of tidal ranges for specified days (-). Stars indicate the maximum daily tidal range for the days on which the 4 cable-breaking flows occurred on January 14–16th, March 9th 2020, April 28–29th 2021, and January 28th 2022, **D** Time series of daily tidal coefficients at river mouth (Soyo) showing times of three cable-breaking turbidity currents (red stars), and peak of major river floods (blue arrows). The larger the tidal coefficient, the greater the tidal range. Period in which ADCP-moorings deployed shown by yellow box.

sediment to become denser than seawater, so that the river-plume plunges to move directly along the seabed as a 'hyperpycnal flow'[27,28,51]. This model can be ruled out for the turbidity currents that flushed the Congo Canyon, because of the significant delay between peak flood discharge and these submarine flows (Fig. 3). The Congo River also has relatively low suspended sediment concentrations, making it unlikely to trigger hyperpycnal flows[52].

However, two other models could explain how floods and spring tides may combine to generate these canyon-flushing flows (Supplementary Fig. 4). In the first model, floods drive large amounts of sand-dominated bedload across the submarine canyon head ('x' in Fig. 1D and Supplementary Fig. 4a). This causes

the canyon-lip to prograde rapidly, and then collapse, forming a powerful turbidity current[30,49]. A significant time delay occurs between flood peaks and all four canyon-flushing flows (Figs. 3 and 4). Thus, although rapidly deposited flood-sediment may prime the canyon-head for failure, it must remain close to failure for weeks to months after the flood, until a minor perturbation sometimes associated with spring tides triggers final failure[30,49,50]. Those perturbations might include expansion of gas bubbles in sediment[53] or increased bedload transport at spring ebb tides[54].

A second model is that major floods supply large amounts of fine-grained mud, which is then stored within the estuary for weeks to months, before being released primarily at spring tides (Supplementary Figs. 4c and 5). This mud is initially dispersed via

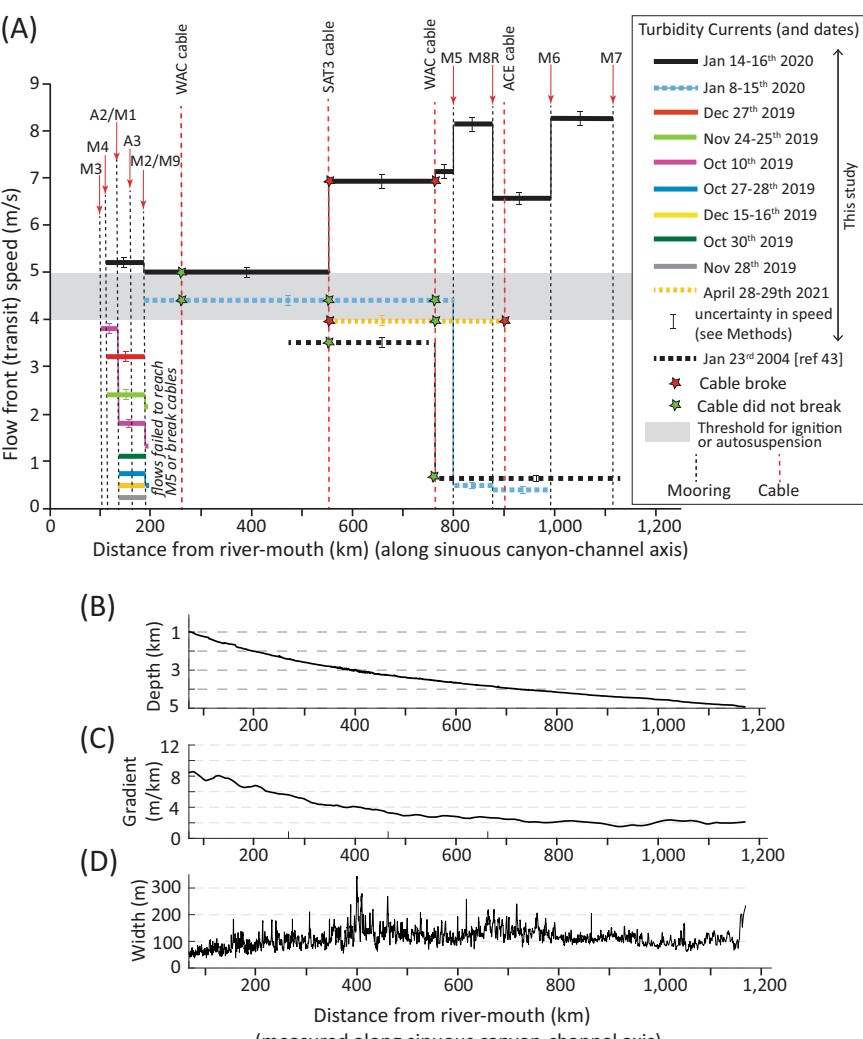

**Fig. 5 Changes in turbidity current front (transit) speed with distance along the Congo Fan system, compared to changes in long profile, gradient and width. A** Changes in front speed with distance from Congo River mouth for all turbidity currents recorded in 2019–20. Flow speeds are derived from submarine cable breaks, and arrival times at moored ADCPs. Distances are measured along the sinuous floor of the Congo Canyon-Channel. Seabed cable are shown by vertical dashed lines, and ADCP-mooring (e.g., M7) sites are shown by red arrows and vertical dashed lines. Speeds of individual turbidity currents in 2019–2020 are shown by different coloured lines. The figure also includes the speed of the April 28–29th 2021 turbidity current between cable breaks (Supplementary Table 1), and the speed of a turbidity current between moorings in 2004[42]. Flows with front speeds >4–5 m/s (grey box) tend to self-accelerate or sustain those front speeds over long distance, while flows with front speeds <4 m/s tend to decelerate and dissipate. **B** Changes in water depth and **C** seafloor gradient with distance along the floor of the canyon-channel. **D** Changes in canyon-channel width with distance measured at crests of confining levees or first terrace.

surface plumes[55] (Fig. 1D), but settles onto the seabed across the entire estuary (Fig. 1D). Field observations (R. Nunny, *pers. comm.*, 2021) from an extensive shallow-water plateau upstream of Soyo (Supplementary Fig. 4a) show that a mud layer accumulates throughout the year (Supplementary Fig. 5). During periods of elevated river discharge, and especially when spring ebb tides also occur, the freshwater plume touches-down across this shallow-water plateau. This causes mud to be resuspended, forming highly-mobile fluid-mud layers[56] that are several metres thick (Supplementary Figs. 4 and 5). These fluid-mud layers then drain into tributary canyon-heads, where they may directly generate turbidity currents, or produce unstable deposits that fail to produce even larger turbidity currents (Supplementary Figs. 4 and 5). Near-bed estuarine circulation may also help to trap fine sediment in this second model[57] (Supplementary Discussion). It is unclear which process(es) generated canyon-flushing turbidity currents, due to lack of observations from the river-mouth.

To understand how turbidity currents transfer sediment from river-mouths to the deep-sea, we also need to understand why some turbidity currents increase in power and runout for exceptional distances into the deep-sea, while other flows terminate in shallow water. It has been theorised that turbidity currents, which erode sediment become denser, and thus accelerate, causing increased erosion, and further acceleration (termed 'ignition'[36]). Alternatively, turbidity currents that deposit sediment decelerate, leading to further deposition ('dissipation'). These positive feedbacks could produce thresholds in behaviour that depend on small differences in initial flow state[36]. It has also been proposed that flows could achieve a near-uniform state in which erosion is balanced by sediment deposition, termed 'autosuspension'[36]. However, it was previously contentious whether ignition or autosuspension were reproduced in relatively slow laboratory-scale turbidity currents, and ignition had not been documented clearly in the field.

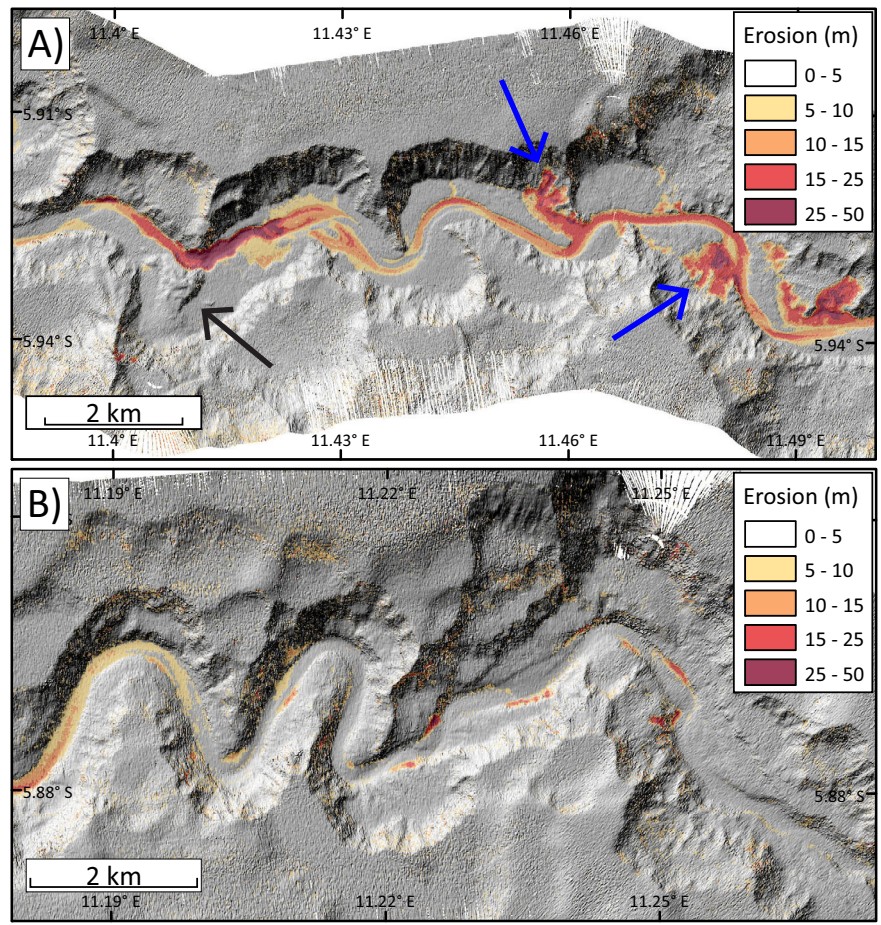

**Fig. 6 Changes in seabed elevation, and patterns of localised seabed erosion, along two sections of the upper Congo Canyon between September–October 2019 and October 2020.** Depth of erosion is shown in red, with (hillshaded) bathymetry from 2019 shown underneath in grey-scale. **A** Location where deep erosion in 2019–2020 is associated with knickpoints (steeper zones in canyon long-profile), between moorings M4 and A2/M1 (see Fig. 1b for mooring locations). Note also the prominent canyon side-wall failures in 2019–2020 (blue arrows), and an older (pre-2019) landslide-dam (black arrow) that created a major knickpoint[64]. **B** Location between moorings A3 and M2/9 (Fig. 1b) where turbidity currents in 2019–2020 caused much less erosion of the canyon floor. Also see Supplementary Figs. 7 and 8 for additional maps showing patterns and magnitudes of seafloor erosion.

This study documents unambiguously that field-scale turbidity currents can ignite, and that ignition can occur over exceptionally long (~1000 km) distances (Fig. 5A). This acceleration cannot be explained by seabed gradients that decrease with distance (Fig. 5C), or canyon-channel width that is broadly uniform with distance (Fig. 5D). However, acceleration is associated with large (2.68 km³ [>1.00 km³]) volumes of erosion (Fig. 6 and Table 1).

Changes in the front speed of turbidity currents with distance have only been measured in detail at five sites[4,24,25,34,35,58,59]. However, three key observations emerge from four locations where flows were confined within canyons-channels (Figs. 7 and 8; Supplementary Discussion and Table 3). First, a common pattern of flow-front speeds occurs. Flows with initial front speeds exceeding ~4 to 5 m/s subsequently runout for longer distances (Figs. 7 and 8). These flow fronts either sustain speeds of 5–8 m/s (autosuspend), or accelerate from ~5 to 8 m/s (ignite). It is these flows that carry the largest amounts of sediment and organic carbon[59], travel furthest, and pose the greatest hazard. Conversely, flows whose fronts travel at <4 m/s tend to decelerate and dissipate. Changes from confined to unconfined flow as turbidity currents exit canyon-channels also cause pronounced deceleration, as in the latter stages of the NW Atlantic event of 1929[4].

Previous theory predicts that sediment grain-size, and thus settling velocity, plays a key role in determining whether a turbidity current ignites or dissipates[35,36]. Thus, a notable result is that similar threshold initial front speeds (4–5 m/s) for ignition are observed in locations with very different grain-size distributions (Fig. 7). Congo Canyon is fed by a muddy river[37], and the upper-canyon floor is mud-dominated[32], while at the other end of the spectrum, Monterey Canyon is fed via sand-dominated long-shore drift[34,35] and has a sandy floor[34] (Supplementary Discussion). It thus appears grain-size is a weak control on front speeds needed for ignition. Previous theories for ignition are based on energy balances or series of equations[36] that often assume flows are relatively dilute (« 10% sediment volume), such that sediment grains settle individually. An alternative model is proposed here (also see ref. [35]) in which faster turbidity current fronts comprise a dense (>20–40% volume) near-bed layer, in which grains do not settle individually, and which is weakly turbulent. Field evidence from Congo Canyon and elsewhere suggests faster turbidity currents contain such a dense near-bed layer at their front, while slower moving flows lack a dense layer[33,34,59]. Behaviour of this dense layer may depend on variations in excess pore pressures, dense layer thickness, substrate properties and erosion rates[60,61], rather than settling velocity of individual grains. Indeed, experiments have shown substrate character and erosion processes can determine if a dense flow grows and accelerates[61].

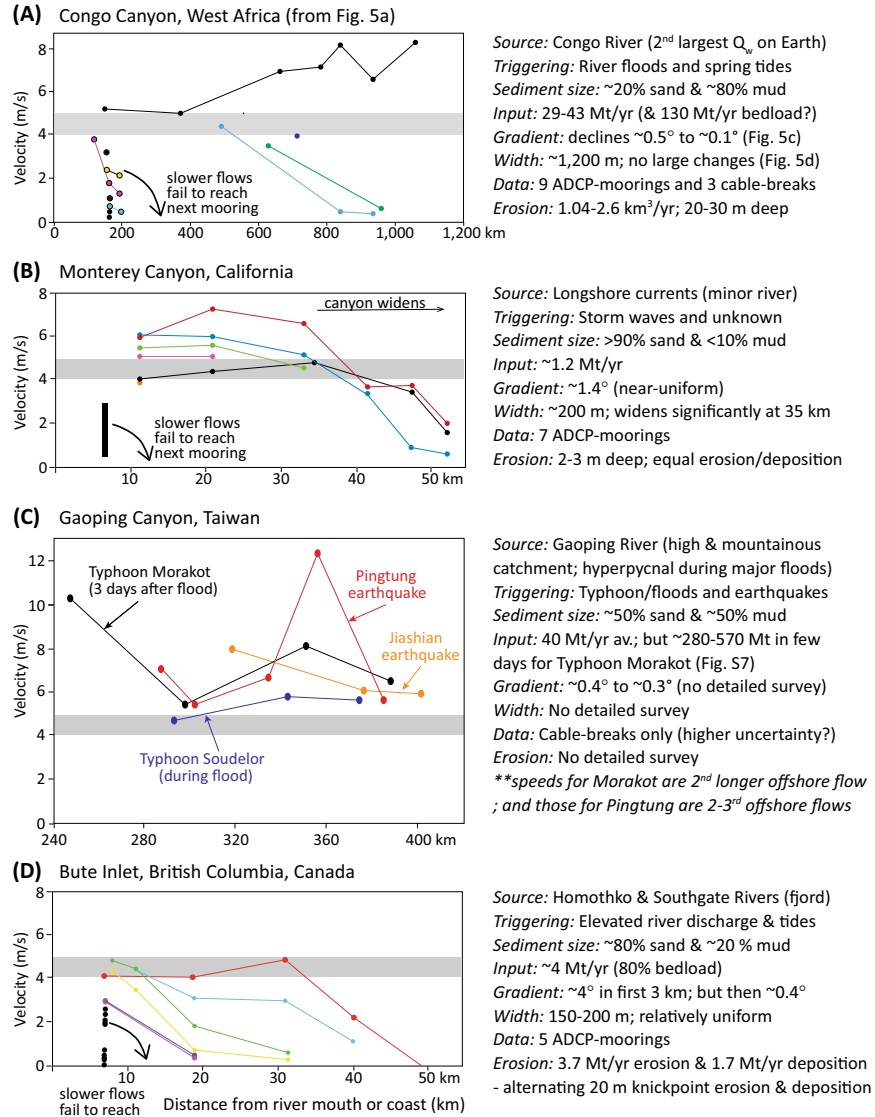

**Fig. 7 Changes in turbidity current front velocity with distance for flows that are confined in canyon and channels, from the four locations worldwide where such data are available.** The threshold flow-front velocity (4–5 m/s) needed for ignition or autosuspension (grey rectangle) is similar in each system, despite major differences in grain sizes, sediment input, triggers and other parameters. These field sites are **A** the Congo Canyon-Channel (this study), **B** Monterey Canyon offshore California[34,35], **C** Gaoping Canyon offshore Taiwan[24,25], and **D** Bute Inlet in British Columbia, Canada[58,59]. Data from individual flows are shown by different coloured lines and dots. Front velocities are averages between moorings or cable-breaks, and distances are measured from the coast or main river mouth. Speeds from Monterey Canyon also include maximum (internal) flow speed measured by ADCPs at the closest mooring to shore (black rectangle)[34,35]. Further information is provided for each system on the source and amount of sediment supplied, fraction of sand and mud, how flows are triggered, changes in seabed gradient and canyon-channel width, and annual volumes or depths of seabed erosion/deposition caused by these turbidity currents (see Supplementary Discussion for more details and references on which figure is based).

However, although initial front speeds are a good predictor of ignition-autosuspension, they are a poor predictor of runout distance, or depth and volume of erosion (Figs. 7 and 8). Flows with fronts speeds of 5–8 m/s in the Congo Canyon ran out for >1100 km, and eroded to depths of 20–30 m, removing 2.68 [>1.00] km³ of sediment. In contrast, flows in Monterey Canyon with comparable front speeds (5–7.2 m/s) ran out for only ~50 km, causing nearly-equal volumes of erosion and deposition, to depths of just 2–3 m[34] (Supplementary Table 3). This may be due to flow-front speeds being determined by local factors[8]. As flows with similar fronts cause very different magnitudes of erosion, this suggests erosion primarily occurs from parts of the flow ('body') located behind the front. Flows with similar fronts can thus have very different bodies. Differences in erosion

magnitude may also be due to variations in seabed sediment strength, such as between the sand-dominated floor of Monterey Canyon and the much muddier floor of Congo Canyon.

Finally, flow-front speeds sometimes change at relatively slow rates over long distances (Figs. 7 and 8). For example, the front of the January 14–16th flow travelled at 5.0–5.2 m/s in the upper canyon (Fig. 5A), despite eroding a large amount of seabed sediment along this reach of upper canyon (Table 1), while flows in Gaoping and Monterey Canyons sometimes had similar front speeds for 30–100 km[25,34,35] (Fig. 7). This suggests that the fronts of faster moving turbidity currents may tend towards a near-equilibrium state. Similar front speeds in different systems (5–8 m/s; Fig. 7) also suggest that a comparable front state may develop in diverse settings (Fig. 8).

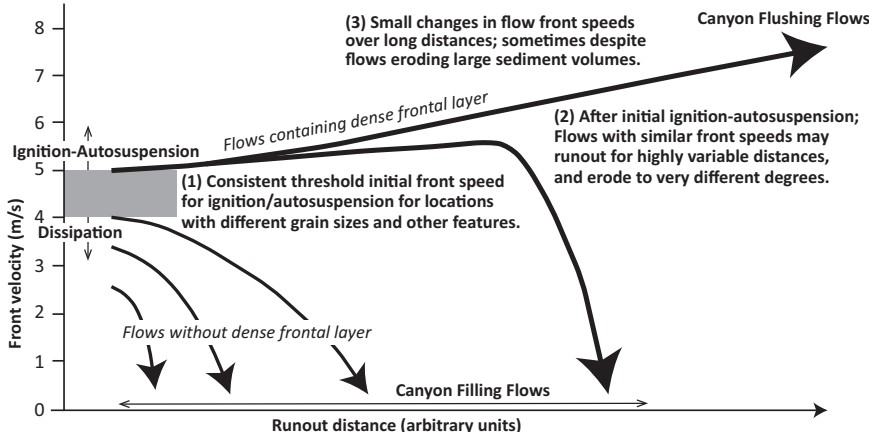

**Fig. 8 Schematic diagram of changes in turbidity current front speed with distance, summarising key observations (labelled 1-to-3) that emerge from comparing field data at four different locations (Fig. 7).** Further details for these locations are provided by Supplementary Table 3. (**1**) There is a consistent threshold of initial front speed (4–5 m/s) needed for ignition or autosuspension in systems with variable sand-mud ratios, trigger mechanisms, input sediment flux, and other factors (Supplementary Table 3). (**2**) After flows have undergone ignition-autosuspension, flows with similar (5–8 m/s) front speeds can then runout for highly variable distances, and erode the seabed to different degrees. Flows in Congo Canyon ran out for >1100 km, and eroded the seabed to depths of 20–30 m (Fig. 6), while flows in Monterey Canyon with similar front speeds (Fig. 7) ran out for 50 km, causing sub-equal amounts of erosion and deposition to depths of only 2–3 m[34]. Flow-front speed can be a poor predictor of final runout distance and seabed erosion, suggesting flows with similar fonts may have different bodies. (**3**) Flow-front speeds can have small fractional changes over long distances, despite sometimes eroding large sediment volumes, as seen for flows in Congo Canyon in 2020 (Fig. 5A; Table 1). This suggests that after ignition-autosuspension, some flow fronts can approach a near-equilibrium state, potentially linked to presence of dense near-bed layers.

We now present a new generalised model for how turbidity currents transfer globally significant volumes of sediment from a major river to the deep-sea (Fig. 9). Previous studies suggested that frequent and smaller turbidity currents deposit sediment within canyons, which are then flushed by much more infrequent and powerful flows. Some studies suggested that flushing flows occurred every few thousand years, and are most likely triggered by earthquakes[4,20,45–47]. Here we show that numerous smaller flows infill the Congo Canyon; indeed these flows are active for 30% of the time in the upper canyon[31–33] (Fig. 2A). Far more powerful and infrequent flushing events then excavate very large volumes (e.g., ~2.68 km³) of sediment from the canyon-channel floor (Fig. 6). However, contrary to some previous models[45–47], this study shows canyon-flushing events can be triggered by floods as well as earthquakes, with clusters of canyon-flushing events occurring after one or more major floods over a period of weeks to months, and possibly years. Recurrence intervals for these major floods is 20–50 years[43], while previous work documented flushing events with recurrence intervals of hundreds to thousands of years[45–47,62]. The sediment mass carried into the deep-sea by a flushing event is comparable to that supplied by the Congo River between flushing events. The Congo River supplies ~29–43 Mt of sediment each year[6,44,55], so the sediment mass (1338–2675 Mt) excavated by 2019–2020 flows is comparable to suspended sediment supply from the river over the last ~31–92 years (Table 1). Thus, although sediment is mainly stored for up to several decades in the canyon-channel floor, it is then efficiently flushed beyond the canyon-channel (Fig. 9).

This new understanding of how river mouths are connected to the deep-sea by turbidity currents (Fig. 9) explains why organic carbon transfer and burial can be highly efficient[15,17]. Fresh organic carbon from major floods can reside in the river-mouth for weeks or months before being flushed into the deep-sea, together with a far larger volume of organic carbon from canyon-floor deposits that accumulated over several decades. The supply of organic carbon by turbidity currents can also have profound impacts on seabed life. For example, distinctive chemosynthesis-

based ecosystems occur on the lobe fed by the canyon-channel, where sediments rich in (mainly terrestrial) organic carbon are rapidly buried[18,19]. This study illustrates how large amounts of organic matter-rich sediment are delivered episodically to this lobe. It also emphasises how turbidity currents physically disturb benthic fauna, as tens of metres of sediment may be removed locally along the canyon-channel floor, sometimes with related side-wall failures (Fig. 6).

Seabed telecommunications cables now carry >99% of global data, underpinning daily lives[22,23]. Cable routes are generally chosen to avoid submarine canyons, but this is not always possible. Cable-breaking flows in this study are sometimes associated with exceptional floods, and such floods could provide an early warning of elevated risks to cables. Elevated risk may persist for a significant period after the flood peak, and a single major flood can generate multiple cable-breaking flows (Fig. 9). A key decision for cable routing is how far offshore the cable should be located from the river-mouth. Turbidity current frequency decreases strongly with distance, as initially slower events dissipate within the upper canyon. However, some larger and more infrequent flows can accelerate and ignite (Fig. 5), causing an increased hazard to cables located further offshore, as they will experience the fastest flow-front speeds.

This study indicates turbidity currents with frontal speeds exceeding 5.5–6 m/s (Fig. 5A) are needed to damage cables, and this is broadly consistent with information from cable breaks elsewhere[25] (Fig. 7). However, although some cables broke in the January and March 2020, April 2021 and January 2022 flows, other cables survived despite being impacted by turbidity currents with similar front speeds (Figs. 2 and 5; Supplementary Table 1). Thus, local conditions can prevent a cable from breaking, while neighbouring cables break. This suggests there may be ways to route cables in more advantageous positions to reduce cable breaks. Time-lapse surveys may provide an explanation for why some cables break, while others do not. These surveys show that seabed erosion during turbidity currents is very patchy, over distances of just a few kilometres (Fig. 6). In particular, deep (20–40 m) erosion may be associated with knickpoints[58,63,64],

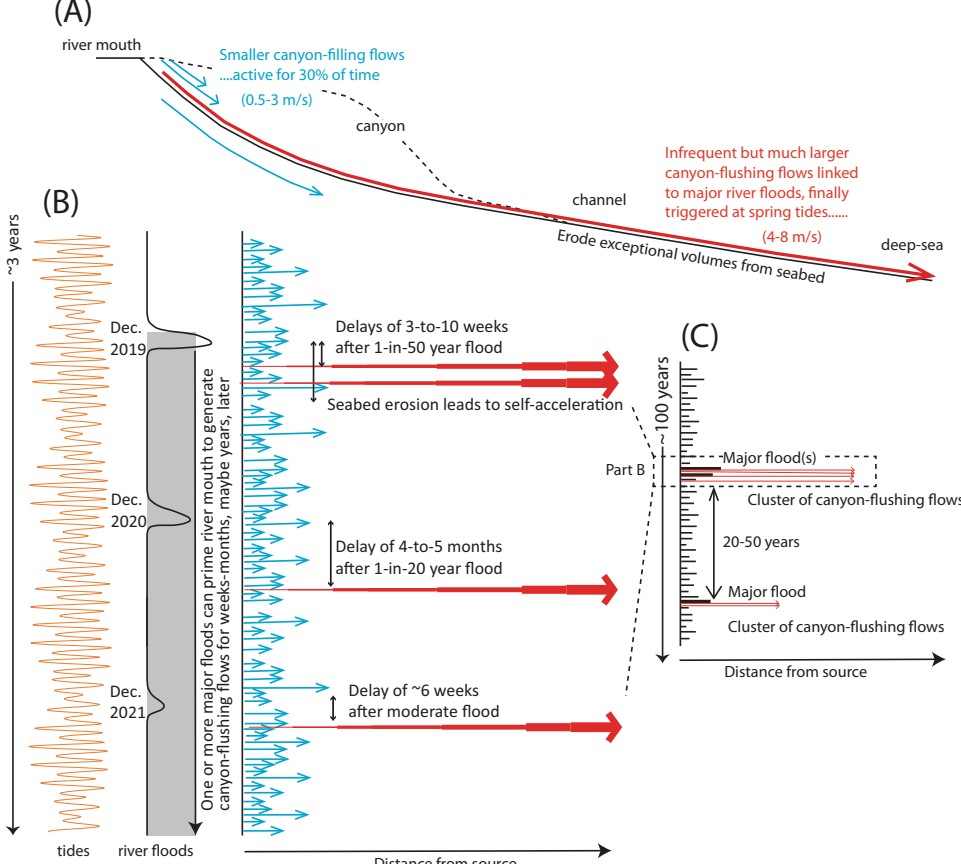

**Fig. 9 Generalised model for how the turbidity current pump operates from river mouths to the deep sea, showing flow timing and frequency, and spatial behaviour and evolution. A** Schematic profile along a generalised submarine canyon-channel from the river mouth to deep-sea. Numerous smaller-scale turbidity currents that infill the canyon (in blue). Much more infrequent, powerful and longer runout turbidity currents then erode the sediment infill from the smaller flows, and flush the canyon (in red). **B** and **C** Time series (vertical axis) showing a sequence of smaller canyon filling flows (in blue) and larger canyon-flushing flows (in red), based on this study of the Congo system. Part B shows canyon filling and flushing flows over a ~3 year period, together with river floods and tidal cycles. Canyon-flushing flows occur 2 weeks to 5 months after major floods (Fig. 3), and coincide with spring tides (Fig. 4). Part C shows a longer 100 year period in with canyon-flushing flows are associated with major floods occurring every 20–50 years.

defined as zones of locally steeper gradients along the canyon or channel floor (Fig. 6), and such localised deep erosion will undermine cables and cause breaks[9].

It has previously been suggested that turbidity current deposits (turbidites) may provide a record of major floods[27,51], which could be valuable if it goes further back in time than records on land. Here we confirm that major river floods can indeed be recorded by deep-sea deposits, although one major flood can generate multiple turbidity current deposits. The best submarine record of major floods is located near the end of the canyon-channel system, as smaller-scale turbidity currents complicate flood-records closer to the river-mouth (Fig. 9).

This study provides the clearest evidence yet that river floods can directly and rapidly impact the deep-sea[27,28,51]. Climate change is predicted to produce a more active hydrological cycle, with global changes to flood frequencies[65]. Future changes in Congo River discharge are uncertain but potentially significant[66]. Here, we show how such changes in terrestrial climate and river-flood frequency may affect how organic carbon is flushed into the deep-sea, associated functioning of deep-sea flood webs, and hazards faced by seafloor cables. Dam construction, deforestation and changes in land-use can also substantially affect sediment flux to river-mouths[6,44,67,68], and this too may change the frequency of turbidity currents. This study of the longest runout sediment flow yet measured in action thus illustrates why changes

affecting terrestrial continents may also have significant impacts on the deep-seafloor.

## Methods

**Field deployment of moorings.** Eleven moorings with ADCPs were deployed (Supplementary Fig. 1) at points along the floor of the Congo Canyon-Channel[9] (Fig. 1), with locations confirmed to within +/− ~15 m by ultra-short baseline acoustic positioning. Three moorings were damaged by smaller flows in the upper canyon, and surfaced before a much larger turbidity current occurred on January 14–16th 2020 (Fig. 2). The remaining eight moorings surfaced on January 14–16th due to this exceptionally powerful cable-breaking flow (Fig. 2). Nine of the 11 moorings were then eventually recovered via emergency vessel charters.

**Arrival times of turbidity currents at moorings and cables.** The arrival times of turbidity currents at ADCP-moorings were defined using the time series of velocity profiles recorded by 75, 300, and 600 kHz ADCPs every 11-to-45 seconds (Supplementary Table 2). The arrival times of turbidity currents were marked by an abrupt increase in near-bed velocities above ambient values of ~0.3 m/s. The timing of faults on submarine telecommunication cables were also used to define turbidity current arrival times (Supplementary Table 1), and this assumes the cables were damaged by the arrival of the flow-front. Cable breaks were recorded to the nearest minute.

**Flow-front (transit) speeds between moorings or cables.** The speed of the flow-front between moorings or cables was calculated by dividing the distance between sites and the difference in arrival times. Distances were measured along the floor of the canyon-channel using bathymetric survey data. Uncertainties in front speeds (Fig. 5A and Supplementary Table 2) arose for following reasons. First, ADCPs

recorded velocity profiles every 9–12 s (moorings M1-M8) or 45 s (mooring A2 and A3), while clock drift for the ADCPs was negligible (<225 s in 6 months; <0.0014% of times). The timing of cable breaks is known to the nearest minute. The main uncertainty in front speed is typically due to <2% uncertainty in distances measured between moorings, which arises from operator choice as to exact location of the deepest part of the canyon-channel floor. The only exceptions are front speeds from moorings M2 to M9, located just ~370 m apart, where the frequency of ADCP measurements becomes important, leading to larger percentage error bars. It is also noted that front speeds from cable breaks assume that the cable is immediately broken by the arrival of the flow's front, while those from moorings assume its position was not changed by previous flows.

**Time at which turbidity currents are triggered**. The first mooring is located ~100 km from the river mouth (Fig. 1). It was thus assumed that turbidity currents originated at the mouth of the Congo River, and that the flow speed from the river mouth to the first mooring was the same as that between the first and second moorings. For faster moving turbidity currents with speeds over 2–3 m/s between the first two moorings, the uncertainty of when the flow originated is likely to be less than a few hours (i.e., the time taken for the flow to travel 80 km at speeds of >4 m/s). Thus, although the original times of these turbidity currents cannot be reliably compared to individual low and high tides, those times can be compared to longer-term cycles of spring and neap tides. Uncertainties in the time taken by flows to travel from the river mouth to the first mooring site are much larger for slow moving flows, and may be several days for flows travelling at <1 m/s (and see Supplementary Information). Thus, it is more challenging to determine if these slower moving flows are also triggered by spring-neap tidal cycles, and they too cannot be linked to individual low or high tides.

**River discharge**. The timing of turbidity currents was compared to fluctuations in water discharge from the Congo River at the Kinshasa gauging station (Fig. 3), located ~400 km from the river mouth, as measured by the Règie des Voies Fluviales (RVF) at Kinshasa, Democratic Republic of Congo.

**Tidal elevations at the river mouth**. Daily tidal data (Fig. 4) were obtained for Santo Antonio do Zaire near the port of Soyo, at the Congo River mouth (Fig. 1A).

**Time-lapse seafloor surveys and eroded volumes**. Swath multibeam surveys of seafloor bathymetry were collected in September–October 2019 and October 2020 using a Kongsberg EM122 (1° x 1°) system operating at 12 kHz for two areas (Fig. 1A). Highest resolution data was generated by setting the swath width to the narrowest setting (45° from the nadir), and having large overlaps between adjacent swaths. Sound velocity profiles (SVPs) were taken through the water column at the start of most surveys, and a second SVP was performed halfway through some longer survey. The first area of repeat surveys was along the upper canyon in Angolan waters, while the second area was the deeper-water channel in international waters (Fig. 1A).

Multibeam sonar bathymetric data were processed in CARIS HIPS and SIPS and corrected for the ship's motion and for differences in sound velocity in the water column (using SVP data), before being gridded with a horizontal grid cell dimension of 5 m (upper canyon in Angolan waters) or 15 m (deep-water channel within international waters). Data were cleaned manually for obvious outliers in CARIS. A bathymetric difference map was then produced by subtracting October 2020 bathymetric data from September–October 2019 bathymetric data.

**Volume and mass of seabed sediment eroded along the Congo Canyon-Channel in 2019–20**. Patterns and volumes of seabed erosion along the Congo Canyon-channel were determined using the 2019 and 2020 swath multibeam surveys (Fig. 1). Changes in elevation were multiplied by grid cell areas to derive volume. Volumes of seabed change did not include the lobe, beyond the end of the deep-sea channel.

Four methods were trialled to determine volumes of seabed change (Supplementary Figs. 7–10). The first three methods define a 'limit of detection' for real seabed change, and values below this limit are then discarded when calculating eroded or deposited seabed volumes. Importantly, these three methods produce volumes of seabed change that are minimum values. The limit of detection may either be spatially uniform (at least within each of the upper-canyon or lower-channel survey areas), or spatially varying with a unique value being assigned to each grid cell[69]. A fourth method assumes that measurement errors are symmetrically distributed about zero, and these errors will thus cancel out over the survey areas. This final method thus returns a 'best guess' for volume of seabed change, rather than a minimum value.

*Method 1*. Changes in seabed elevation were measured for areas that are assumed to have undergone no significant («1 m) change from 2019–20. These areas were located outside the main canyon-channel axis (Supplementary Figs. 7 and 8). Histograms of seabed changes in these areas (Supplementary Figs. 7 and 8) are then used to define a 'limit of detection' for real seabed change in other parts of the same survey. This analysis suggests that changes in seabed elevation between surveys of

<4–6 m in the upper canyon, and <10–15 m in the lower channel, can be caused by measurement errors (Supplementary Figs. 7 and 8).

*Method 2*. It is often assumed that uncertainties in seabed elevation for individual surveys are <0.2 % to <0.5% of the water depth[69], which would lead to uncertainties of 3 to 10 m in the upper canyon (~1.5 to 2 km water depth), and 6 to 25 m in the deeper-water (~3 to 5 km) channel (Fig. 1). Uncertainties from both surveys then need to be combined when calculating the limit of detection in seabed change between surveys[69]. Thus, assuming uncertainties from each survey are summed, this method gives a limit of detection for seabed elevation change of 6–20 m in the upper canyon, and 12 to 50 m in the lower channel. Method 1 (Supplementary Figs. 7 and 8) suggests that the lower range of these estimates are most likely, for these surveys with narrow beams. Non-random spatial patterns of seabed change > ~5 m in the upper canyon (Supplementary Fig. 7d, e), and >15 m in the lower channel (Supplementary Figs. 8d, e), which are physically reasonable (e.g., non-random and focussed along the canyon floor only), also suggest a reasonable limit of detection is closer to the lower end of this range estimated by method 2.

*Method 3*. The CUBE algorithm implemented within software (CARIS) typically used to process multibeam echosounder data automatically provides an estimate of spatially varying uncertainties for different grid cells[69]. These CUBE-derived uncertainties include additional important factors, such as whether data come from inner or outer acoustic beams in the multibeam sonar array, and are thus preferable to other methods[69]. CUBE-derived uncertainty values are then combined in quadrature (to ensure all resulting values are positive) to derive combined uncertainties in elevation changes between two surveys[69]. CUBE-derived uncertainties in changes in seabed elevation are typically less than 5 m in the upper canyon (Supplementary Fig. 9b, c) or 10–15 m in the lower channel (Supplementary Fig. 10b, c). The upper limits of the spatially variable CUBE-derived uncertainties are thus also broadly comparable to those derived via to Method 1. The CUBE-derived uncertainty value for each grid cell in the difference map can then be multiplied by a constant termed $k$[69]. A value of $k = 1$ ensures that raw uncertainties values calculated by CUBE are used as the limit for detection, while higher values of $k$ are more conservative. Mountjoy et al.[20] used a value of $k = 1.96$ (two standard deviations or 95% confidence limits) to define a limit for detection[69]. Higher values of $k$ generate progressively more conservative limits of detection, and provide greater confidence that seabed change is real. However, they also cause data from more grid cells to be discarded. Indeed, a sufficiently high value of k will conclude with ~100% percent confidence that at least zero seabed change occurred; which is not a useful conclusion[69].

*Method 4*. A final method assumes that measurement errors are symmetrically distributed about a zero value, and these errors will thus cancel out over the survey areas. This final method thus returns a 'best guess' for volume of seabed change, rather than a minimum value.

*Error bars for volumes of seabed change*. Methods 1–3 are based on limit(s) of detection below which measured values of seabed change are discarded when calculating volumes of change. These limits of detection can produce error bars for volume of seabed change, via multiplying the area of the grid cell and the corresponding limit of detection. However, this approach can produce very large ranges for error bars, which may indeed exceed the main estimate of seabed volume change[69]. Importantly, such an approach implicitly assumes that errors can reach maximum values simultaneously at every grid cell, and that errors are thus not close to being symmetrically distributed about zero. Thus, and as noted by Schimel et al.[69], the significance of error bars derived by multiplying limit(s) of detection by grid cell area is thus uncertain.

*Chosen method*. In this paper we follow the method of Mountjoy et al.[20] for reporting volumes of seabed change, and take into account some recommendations of Schimel et al.[69]. We report volume changes in the format X [>Y], where X is a 'best guess' that is simply based on changes in seabed elevation measured at all grid cells (i.e., Method 4). This 'best estimate' assumes that measurement errors are close to being symmetrically about zero, and thus cancel out. However, we also then report a minimum estimate for volume of seabed change (Y), which is based on a limit of detection. We choose to use the CUBE-derived uncertainties in seabed elevation change at each grid cell, as they incorporate a wider range of important uncertainties than other methods[69]. We also chose a value of $k = 1.96$ (Method 3) following Mountjoy et al.[20], and note that this is a rather conservative limit of detection[69]. In Supplementary Figs. 9 and 10 we therefore also show the volumes of seabed change derived using all four methods, comprising (i) no limit of detection and using all values of seabed change in every grid cell, spatially variable limits of detection based on CUBE-derived uncertainties and a value of (ii) $k = 1$ and (iii) $k = 1.96$, and (iv) a spatially uniform limit of detection that is 5 m in the upper canyon and 15 m in the deep-water channel. The volume of seabed change is also split into volumes of erosion and deposition (Supplementary Figs. 9 and 10), although deposited volumes are often much smaller. This shows how the different methods affect (sometimes minimum) volumes calculate for erosion, deposition and net change (Supplementary Table 4).

**Conversion of eroded volumes to mass**. The repeat surveys in 2019 and 2020 only covered 40% (477 km of 1179 km) of entire length of the Congo Canyon-Channel, as measured along its sinuous axis. This includes a 112 km survey from water depths of ~1.6 to 2 km in upper canyon (9% of total length) and a 477 km (31% of total length) surveyed of the deep-water channel (>3.3 km) to its termination. It was estimated that 1.07 km$^3$ [>0.40 km$^3$] of sediment was eroded from 40% of the entire system length. It is reasonable to assume similar rates of erosion occurred within the intervening stretch (Fig. 1), and it is therefore estimated that 2.68 km$^3$ [>1.00 km$^3$] of seabed sediment was eroded along the whole canyon-channel system (Supplementary Table 4).

Volumes of eroded sediment (in km$^3$) along the Congo Canyon-channel were converted to sediment dry mass (Mt) in the following way, to allow comparison to other global sediment mass fluxes (Table 1). An average porosity of 60–80% was assumed in the eroded sediment volume based on global data for the upper 50 m of sediment[70], which was filled with seawater. A sediment grain density was assumed of ~2.5 g/cm$^3$, which is somewhat less than the 2.6 g/cm$^3$ density of quartz, in order to account for less dense grains (e.g., ~2–3% of organic matter), and a seawater density of 1035 g/cm$^3$. This implies a wet sediment density of 1.33–1.62 g/cm$^3$, and dry sediment density of 0.5-1 g/cm$^3$. This is consistent with wet sediment density seen (1.1 to 1.6 g/cm$^3$) in cores through the upper few metres of sediment in the Congo Fan, while noting that sediment density will increase below those upper few metres below the seabed[70], and seafloor erosion often reached depths of 20–30 m (Fig. 6).

## Data availability

Data on flow arrival times supporting the findings of this study are available in the Supplementary Data files. The 2019 and 2020 swath multibeam survey data, and the resulting map of changes in seabed elevation from 2019 to 2020, that support the calculation of eroded volumes are available to download via the British Oceanographic Data Centre (BODC) [https://doi.org/10.5285/dfe7a980-89d8-2830-e053-17d1a68b81ba].

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

## Acknowledgements

We are especially grateful to those who helped to overcome a series of major challenges to recover moorings and data during a global pandemic, after the moorings unexpectedly surfaced. They include Jez Evans, Natalie Powney, Guy Dale-Smith, Colin Day, Eleanor Darlington, Paul Provost, Mark Maltby of the UK National Marine Facilities, and the officers and crew of the RRS James Cook. Michel Senechal (Orange Marine), and Master and crew of the RV Thevenin recovered a series of moorings. Alan Ainsworth, Master and crew of the Thor Frigg, who recovered an initial mooring, together with Aymeric Frisch Master and crew of the Maria Francesca, who recovered a final mooring, while efforts of Nicholas Ravalac in Point Noire were also appreciated. We thank those who arranged the contract for mooring recovery at short notice, including at the University of Hull, University of Durham (Leila Cole et al.), and NERC. We are also grateful to Max Boyce (Alacatel Submarine Networks), Takalani Tshivhase (MTN Group), and others for conversations about ongoing cable projects. We are also grateful to Gabriel M. Mokango (Technical Director of Regie des Voies Fluviales in Kinshasa) for access to gauging data for the Congo River, while Joao Baptista helped greatly with initial planning. Research was funded by UK National Environment Research Council (NERC) grants NE/R001952/1 and NE/S010068/1 led by P.J.T., by a Royal Society Dorothy Hodgkin Fellowship (DHF\R1\180166) to M.J.B.C., and by a Leverhulme Trust Early Career Fellowship (ECF-2018-267) to E.P. River flow analysis was funded by the Royal Society Africa Capacity Building Initiative grants AQ150005, and FLR\R1\192057, and FCG \R1\201027 to M.A.T., R.T. and G.B. M.C. acknowledges support from NERC COP26 Adaption and Resilience Programme, NERC Climate Linked Atlantic Sector Science (CLASS) National Capability Programme (NE/R015953/1), and the International Cable Protection Committee.

## Author contributions

P.J.T. wrote the manuscript, assisted by M.B., E.P. and S.C.R., and with comments from all of the authors. P.J.T., M.L.B., E.P., R.S.J., M.S.H., S.H., S.M.S., M.H., C.J.H., S.C.R., C.M., R.A. and A.F. collected data on research cruises in 2019 and 2020. M.J.B.C., D.P., M.A.C., M.U. helped to design the field experiment. D.W. and A.G. assisted with cable fault data. R.S., M.A.T., G.B., R.N. provided information on river discharge and processes. C.C. and R.F. led the Angola project component. A.G. and C.P. assisted with bathymetric survey, while R.B. and J.N. assisted with data analysis.

## Competing interests

The authors declare no competing interests.
