## [Peer Review File · Nature Communications]

REVIEWER COMMENTS

Reviewer #1 (Remarks to the Author):

Dear Senior Editor Dr. Sebastian Mueller,

Thank you very much for inviting me to review this extremely interesting manuscript of Talling and co-authors. This manuscript builds up on a very ambitious and dense monitoring experiment of the Congo river-channel-canyon system. This impressive project represents some high-risk science as it aims to directly measure sediment-density flows in this system and it is not a given that large, long run-out flows actually occur during the monitoring deployment phase.

Fortunately, the authors have been able to capture several smaller-scale turbidity current events followed by two large run out events in early 2020, that actually damaged part of the monitoring and seafloor cable infrastructure. This is one of these beautiful & lucky coincidences that advance our scientific knowledge significantly. Hence, the data set represented in this manuscript is absolutely exceptional and one of its kind. It represents the first documentation of these highly erosional, long run-out and system-flushing sediment transport events that seem to be triggered by river floods in conjunction with tidal activity.

Based on these unprecedented measurements, the authors present two novel main concepts:

1. new & detailed conceptual models that explain the generation of such large flows by river floods and tidal activity
2. the comparison with frontal flow speeds and run-out distances from other sites around the world indicates that there is a threshold in initial flow frontal speeds around 4-5 m/s that determines whether flows ignite or whether they 'die out' soon after initiation.

These findings are immensely important to enhance our understanding of the timing and magnitude of the transfer of sediment organic carbon and associated pollutants (e.g., microplastic) into the deep sea as well as the importance of river flooding and climate change on this material flux.

As you may sense from my excitement, I think this manuscript is highly suitable for 'Nature Communications' and I strongly suggest its publication.

The manuscript is clearly structured and well written, the resulting models are clearly explained and illustrated in the Supplementary Material.

I have some comments and suggestions mainly on the representation of the data, data errors and sediment volume calculations. All detailed main and minor comments and suggestions are amended to the two word documents in review mode, so I will spare you a long list of minor comments here. This manuscript – in my view – can be published after minor revisions.

Main Comments:

1. Sediment volume comparison

As the sediment volume comparison is so central to this manuscript & appears very prominently in the abstract, a bit more detail is needed to discuss this comparison:

The value of global annual sediment by Milliman and Farnsworth is $19\,000 \times 10^6$ t/yr. To convert this into km³, the authors seem to use a density of 2.6 g/cm³ (also indicated in Table 1). This is certainly a standard procedure and a good approximation for siliciclastic sediment. But should

lower densities (organic matter, some clays etc) also be considered? If so, then the total global annual sediment load in km³ (converted from t/yr) increases. To my knowledge the data reported by Milliman & Farnsworth generally only comprise suspended river sediment load and these estimates do not contain bedload (as it is notoriously complicated to measure). Hence, the total sediment volume is definitely higher than this value.

This value is then compared to the sediment volume removed by erosion from time-lapse bathymetry maps from the Congo Canyon/ Channel floor. These volumes contain the porosity of the sediment which can reach up to several 10s % at the sea floor.

On a similar note: Time-lapse bathymetry and the resulting difference maps can result in significant errors of estimated eroded volumes. Please indicate the error ranges that these are associated with and explain your error calculation in the Methods section. As you are claiming that these very large sediment volumes are redistributed in just a single year, the errors on that estimate (and therefore possible minimum and maximum volumes) are important.

To make this a fair comparison, these factors should be mentioned and corrected for as much as possible. For now, I think the eroded volume by turbidity currents making up "14-28% of the annual global (suspended?!) sediment flux from rivers" is a maximum estimate that may need some down tuning. However, eroded volumes by turbidity currents in the Congo are very large!

2. Representation of frontal speed measurements

And in order to assess the speed of ignition and a possible steady state velocity, it is important to report the error bars and variability in these measurements. Please add these in the main text, methods and figures. Frontal speeds of the turbidity currents are estimated by the arrival times at the mooring & distance in between moorings. But how do these velocities compare to the actual measured ADCP velocity profiles? Do they agree?

3. Threshold initial front speeds & grain size

The authors pose the new working hypothesis that threshold initial front speeds (for ignition) are not strongly depended on grain sizes. This is a very interesting finding that needs further testing. However, it indicates that these huge muddy and carbon rich systems (e.g., the Congo system) can be equally efficient in transporting material to the deep sea as sand-dominated systems. In order to strengthen this point, it may be useful to incorporate actual grain sizes measurement into e.g., figure 6.

4. Estuarine turbidity maximum

The exact nature and sediment-trapping process by an "estuarine turbidity maximum" remains somewhat unclear in the explanation and the supplementary figure 4d. Can you expand on the explanation and it's illustration in the figure here to clarify?

5. Use of colour scales

The (mis-) use of colour maps that visually distort data through uneven colour gradients or are not readable by people with colour-vision deficiency has been a matter of ongoing discussion (e.g., Crameri et al., 2020). The rainbow colour scale is one of these non-scientific colour scales. I recommend to get rid of the rainbow and use a scientific colour scale (e.g., from this library <https://www.fabiocrameri.ch/colourmaps/>) in all figures of the main manuscript and the supplement (see word documents for details.)

It has been a pleasure to review this manuscript.

**Best Wishes,
Anne Bernhardt
(Freie Universität Berlin)**

Reviewer #2 (Remarks to the Author):

This is a very important paper. It is the first along flow monitoring of the sort of scale of turbidity current that deposits turbidites of significant size in oceanic depths, in contrast to the tiny flows at Monterey, Squamish and elsewhere. It is a remarkable technical achievement. It demonstrates clearly (a) that priming of large sediment accumulations by flood supply leads to initiation of turbidity currents weeks to months after the flood; and thus more generally provides a convincing mechanism for the river to turbidity current transition which to date has been lacking for large muddy rivers. (b) It shows that some flows accelerate by ignition and rework sediment from the canyon floor. This latter process has been (some would argue speculatively) inferred from ancient and historical flows, but never previously demonstrated in action. It is the process that accounts for turbidite beds decimetres to metres in thickness that can be studied in ancient rock successions.

I have no significant disagreements with the data interpretation. The authors are cautious in claiming only what is clear from their data. I have commented in places on the main text.

The paper is irritating to read because of the seeming obsession of being the first to do something. Pointing it out once is useful, but it would be better to emphasize the scientific significance of the findings. The paper is a little repetitious between the front end and the back end.

It would be useful if the term "sediment" were qualified. What size sediment is delivered to the Congo mouth (x% fine sand, y% mud). What is the sediment type reworked from the floor of the canyon – sand or mud? What is the sediment type deposited on the lobe. Older turbidite literature often claims differences between sandy flows (Squamish, Monterey) and muddy flows (?Congo, Bengal) in flow characteristics and architecture. The brief statement on line 294 should have come earlier and should have been a bit more quantitative. It could be summarized in part from the Supplementary Material referencing the more detailed discussion there.

The authors seem to place great significance on discovering that large flushing events are not triggered by earthquakes. Perhaps the literature is heavily weighted towards earthquake triggering of turbidity currents, but it is not surprising that large canyon flushing flows have multiple origins. It is what has been shown by Bailey et al. in Monterey canyon, inferred from core data at Squamish. And was pretty clear in La Jolla canyon and fan in 1970. I recall a Gaoping canyon study which had a tc several days after the flood peak. It would have been useful to point out the analogy with Squamish in the spring tide influence on sediment instability (although the process may have been different, e.g. pore pressure as in the Fraser delta).

In lines 354-357, there is some speculation on patchy erosion and knickpoints. This sounds like the evidence for "cyclic steps", reflected in bedforms at Squamish. Should the authors comment

on this possibility.

I have no specific comments on the Supplementary Material, except to note that a little of what is covered there in the text could usefully be moved to the main paper. The space for this could be achieved by a bit of tightening of repetition and claims of priority.

Well done!

David J.W. Piper

david.piper@canada.ca

Dear Reviewers,

We would like to thank both reviewers (and the editor) for a particularly detailed set of comments that are both constructive and challenging. Our response is below (in blue), and caused us to rethink and revise sections of the manuscript. Such detailed feedback is especially appreciated at our end.

General Comments

We are glad that the first reviewer felt the data was ‘absolutely exceptional’, and that ‘this is the first documentation of these highly erosional, long run-out and system-flushing sediment transport events’, such that ‘findings are immensely important to enhance our understanding of the timing and magnitude of the transfer of sediment organic carbon and associated pollutants’; and that the second reviewer states ‘This is a very important paper. It is the first along flow monitoring of the sort of scale of turbidity current that deposits turbidites of significant size in oceanic depths’.

We were asked to take particular care about avoiding strong language, such as using terms like novel and first. Thus, we have toned down the statements of novelty, including as follows:

- We modified the first line of the abstract (just by moving the single word ‘directly’) to say that ‘Here we document directly for the first time how major river floods connect to the deep-sea, by analysing the longest runout sediment flows (of any type) yet measured in action’. This statement is correct, as these are indeed the first detailed, along flow observations for turbidity currents in the deep (> 2 km) sea. Thus they ‘directly’ document sediment transfer from river floods to deep-sea, unlike previous work that was based on ‘indirect’ and more uncertain lines of evidence, such as seabed deposits or geomorphology.
- We entirely removed the following section from the start of discussion (old lines 200-203). ‘Here we discuss the first detailed direct measurements from turbidity currents in the deep (> 2 km) sea. These unique measurements show how sediment can be transferred efficiently from a major river mouth to water depths of ~5 km, by the longest runout sediment flows (of any type) yet measured in action on Earth’.
- The following (old lines 143- 144) section was modified from ‘However, no previous study had deployed ADCP-moorings at multiple sites extending from upper canyon to the deep sea, as occurred during this project in 2019-2020’ to ‘However, no previous study had deployed ADCP-moorings at multiple sites to the end of a deep-sea canyon-channel, as occurred during this project in 2019-2020’. This is because recent flow monitoring in

Monterey Canyon also had multiple moorings, but only to 1.85km depth, and those moorings did not extend to the deep (2-5 km) sea, nor to the end of a canyon-channel.

- We removed 'for the first time' from lines 205 and 206 in the Discussion. It now reads 'Here we document directly that major river floods generate powerful and long runout turbidity currents that flush very large amounts of sediment through submarine canyons'.
- We amended (old) lines 314-315 in the Discussion by adding 'some', so that it reads 'contrary to some previous models'. As correctly noted by the reviewers, there may be over emphasis on earthquake triggers for flushing events, and not all past work invokes them.
- We also toned down the statement about this being the most detailed information yet on how floods are recorded in the deep sea (new lines 86-87).

Main overall changes in response to reviews

We also made a series of changes based on the comments from the two reviews, and the main changes are now briefly summarised, with more detailed responses below. In response to Reviewer 1, we: (1) revised the section on how eroded sediment volumes (km^3) were converted to sediment mass (in Mt), using a realistic porosity range, and assuming the pores to be water filled, and this was a very useful comment. (2) We also undertook a much more thorough analysis of uncertainties in seafloor elevation changes, and now adopt the same method (X [$>$ Y]) used by Mountjoy et al. (2018 in Science Advances) as set out in a detailed new Methods section and Supplementary Figures. Our thanks for the excellent suggestion here too. (3) We also now use the most recent published estimate of Anthropocene global riverine suspended sediment flux (7,200 Mt by Syvitski et al., 2022), and restrict our comparison to riverine suspended sediment load. (4) We provide uncertainties for flow front speeds for Figure 3 and in Supplementary Table 2, which are discussed in the Methods. (5) We provide a detailed discussion of grain sizes in the four systems we compare, by adding summary information to Fig. 7 and Supplementary Table 3. (6) We provided further explanation of the estuarine turbidity maximum model via the supplementary material, and (7) revised the colour-bar scales as suggested.

In response to reviewer 2, we have (8) toned down claims of novelty and use of 'first' (see bullet points above in response to editor); (9) provided more information on grain size in the supplementary material; (10) added further discussion of past literature that hypothesised river floods could trigger turbidity currents that reached the deep sea, including by Reviewer 2, which now forms a new Supplementary Discussion (as space was too tight in the main text). We also note how this past work is based on more indirect evidence (e.g. turbidite deposits) with greater uncertainties, and how it is typically assumes triggering via (hyperpycnal) plunging river floods (that

we do not invoke in this paper). (11) The reviewer is correct that cable breaks offshore Taiwan after typhoon-related floods in 2009 and 2015 provided direct evidence of long runout turbidity currents after floods, and this is now noted in the main text, and forms a more detailed Supplementary Discussion and Figure 6. However, due to a lack of repeat time lapse surveys, it is still unclear whether these events off Taiwan flushed large sediment volumes through their submarine canyon, nor are the timing of flows verified with ADCP-moorings. (12) We shortened the text where there were repetitions as suggested, to make room for other additions. Our thanks to the reviewer for those specific suggestion on where to reduce the length of the paper, which is now 5,000 words.

Flow efficiency: Reviewers' comments on the wider implications of this study for the efficiency of material transport to the deep sea, and whether sandy or muddy flows were equally efficient, were perceptive and important. They made us think further, and we realised that there are actually three main conclusions from the comparison of flow front speeds (old figure 6, new figure 7), not just one. Thus, we amended the text and added a new figure (figure 8) to emphasise that as well as; (1) threshold initial front speed (4-5 m/s) for ignition-autosuspension is weakly affected by grain size, it is also apparent that (2) once flows have ignited-autosuspended, flow front speed is then a rather poor predictor of final runout and degree of seabed erosion. Flows with similar front speeds can travel for very different distances (e.g. 50 km v. 1,100 km), and (3) changes in flow speed are often rather gradual, despite sometimes eroding a (very) large volume of sediment, so flow fronts may tend towards a near-equilibrium (autosuspending) state. We hope that all three of these key messages about flow front speeds are clearer in the revised version, with the new figure. We would prefer to keep that new figure, but if preferred - it could also potentially go in supplementary material. The main body of text is still < 5,000 words, despite addressing the various points above. We also made minor changes to figures 5-8, to help make them clearer and more easily understood.

Additional cable-breaking turbidity current in January 2022: Finally, the original draft was rather overtaken by events, when a fourth powerful and long runout turbidity current broke the SAT-3 and ACE cables on 28th January 2022. Even though this new event complicates the paper, albeit in interesting ways, we felt it was important to include it for completeness. Thus, we have updated figures 3, 4, and 9, and Supplementary Tables, and modified the text in a few selected places. At this short notice we were only able to obtain Congo River discharge data to December 14th 2021, but these data appears to (just) capture the flood peak, and we do not have exact timing of cable breaks to calculate a transit speed. But we think this additional event adds to the impact of the publication.

The timing of the January 28th 2022 event is indeed consistent with the main conclusions of the previous draft. It was also delayed a few weeks after the peak Congo River discharge, and thus resembles the January 2020 event; indeed all of the four cable-breaking flows have delays of weeks to months after floods. However, the January 2022 event occurred after a rather moderate (near average) peak annual river discharge, and it did not occur at a spring tide. Elsewhere, we note that one of the cable breaking turbidity currents offshore Taiwan (after Typhoon Soudelour in 2015) also occurred during a rather small flood, as shown in new Supplementary Fig. 6. Thus, three of the four cable-breaking flows in Congo Canyon are linked to spring tides, especially those with longer delays in March 2020 and April 2021; but one event is not linked to a spring tide. In most cases there is a link to spring tides, but not in all cases. We therefore made a series of changes include the following.

In the abstract we state that 'We show river-floods also generate canyon-flushing flows, primed by rapid sediment-accumulation at the river-mouth, and sometimes triggered by spring tides weeks to months post-flood', and shorten the abstract by a few words elsewhere (to be 150 words long).

The Results section now says (new lines 164-169): *'A fourth flow broke cables on 28th January 2022, ~6 weeks after a modest (54,651 m³) annual peak in Congo River discharge (Fig. 3). There were significant delays between the flood peaks and the cable-breaking flows, and three of the four cable-breaking flows coinciding with subsequent spring tides (Fig. 4). It appears that floods supplied large amounts of sediment that primed the river mouth to produce powerful and long runout flows (Fig. 3), which were triggered finally ~3 weeks to 4.5 months after flood peaks, in some cases at spring tides'*.

The Discussion section now says (new lines 215-233): *Turbidity currents that flushed the Congo Canyon were associated with river floods, and in most cases spring tides (Figs 3 and 4). Recent studies have shown how elevated river discharge and tides can combine to generate much shorter runout (1-50 km) turbidity currents offshore from smaller river mouths^{30,50-52}, and how the threshold suspended sediment concentration of rivers needed for offshore flows is much lower than once thought⁵⁰ (Supplementary Discussion). However, this is the first study to show that floods and tides can generate far larger turbidity currents offshore from one of the world's largest rivers, and in an estuarine setting. This suggests that floods and tides may trigger turbidity currents in an even wider range of settings than previously thought, which then transfer globally significant sediment volumes. Delays of several weeks to months occur between river floods and turbidity currents that flush the Congo Canyon (Fig. 3). Previous work has documented significant delays between river floods and associated turbidity currents, but only for hours⁵¹ to days^{24,25}, not weeks to months. This suggests*

that river-mouths can store flood sediment for up to several months, and maybe years, and thus act as an efficient 'capacitor', before eventually releasing sediment in one or more long-runout turbidity currents.

The January 2022 event occurred after a moderate peak in annual Congo River discharge, and not at a spring tide (Figs & 4). Long runout turbidity currents can therefore also be caused by smaller floods, and this is also shown by cable-breaks off Taiwan in 2015 after Typhoon Soudelour (Supplementary Fig. 6). It is possible that preceding much larger river floods supplied sediment that contributed to generating long runout turbidity currents in later years. Older cable breaks (1883 to 1937) in the Congo Canyon⁹ also indicate that clusters of cable breaks may occur for several years after major floods (Supplementary Fig. 2).

We have also added a short initial Supplementary Discussion section, which discusses in more detail the links between the four flows and river floods and tides, and how there may be a long term effect of unusually large (1 in 20 and 50 year) floods in 2019/2020, given cables had not broken in 18 years. We also made some small changes in the text to acknowledge not all events occurred at spring tides.

Finally, as set out above, we use the same method as Mountjoy et al. (2018) for reporting sediment mass and volume (X [$>Y$]), where X is the 'best estimate' and Y is a 'conservative minimum estimate'. However, there is not enough space in the abstract to explain the X [$>Y$] reporting method, such that its use in the abstract could potentially cause confusion. An alternative strategy would be to just report the 'best estimate' (X) in the abstract, and only the X [$>Y$] reporting method in the main text (after it has been explained at first use). We are very happy to take advice on what is preferred here.

In the following section, we now provide detailed responses to specific points raised.

Reviewer #1 (Anne Bernhardt)

We thank the reviewer for a particularly perceptive and detailed review, which indeed challenged us to tighten up and improve several key aspects of the manuscript. This is very much appreciated.

We are glad the reviewer feels that this data-set is 'absolutely exceptional and one of its kind', and it is 'first [direct] documentation of these highly erosional, long run-out and system-flushing sediment transport events that seem to be triggered by river floods in conjunction with tidal activity'. The project was indeed high risk, and it was extremely challenging to recover the drifting moorings, especially during CoV-19 lockdowns. We hope our acknowledgements do justice to the efforts of many colleagues in recovering those moorings, which is why the acknowledgements are rather long.

The reviewer's comments on the wider implications of this study for the efficiency of material transport to the deep sea, and whether sandy or muddy flows were equally efficient, were very useful and important. The first main conclusion from the study is indeed about how floods and tides combine to cause these canyon-flushing turbidity currents. However, the review made us think further, and realise that there are three main conclusions from the comparison of flow front speeds in four locations worldwide (old figure 6, new figure 7), not just one. Thus, we amended the text and added a new figure (new figure 8) to emphasise that as well as (1) the threshold initial front speed (4-5 m/s) for ignition-autosuspension is weakly affected by grain size, that (2) once flows have ignited-autosuspended, flow front speed is a rather poor predictor of final runout and degree of seabed erosion (i.e. scale of the flow, and thus efficiency of material transport). Flows with similar front speeds can travel for very different distances (e.g. 50 km v. 1,100 km). In addition (3) changes in flow speed are often rather gradual, despite in some cases eroding a very large volume of sediment, so flow fronts may approach an equilibrium/autosuspension state. The second point also has some important follow on implications, such that once ignited, similar flow fronts have very different bodies, and thus that flows front and body are in 'rather poor communication'. We did not have space in the main text to explore these points (as they are wordy to explain), but they may well form a basis for subsequent publications too.

1. Sediment volume comparison

As the sediment volume comparison is so central to this manuscript & appears very prominently in the abstract, a bit more detail is needed to discuss this comparison. The value of global annual sediment by Milliman and Farnsworth is $19\,000 \times 10^6$ t/yr. To convert this into km³, the authors seem to use a density of 2.6 g/cm³ (also indicated in Table 1). This is certainly a standard procedure and a good approximation for siliciclastic sediment. But should lower densities (organic matter, some clays etc) also be considered? If so, then the total global annual sediment load in km³ (converted from t/yr) increases. To my knowledge the data reported by Milliman & Farnsworth generally only comprise suspended river sediment load and these estimates do not contain bedload (as it is notoriously complicated to measure). Hence, the total sediment volume is definitely higher than this value. This value is then compared to the sediment volume removed by erosion from time-lapse bathymetry maps from the Congo Canyon/ Channel floor. These volumes contain the porosity of the sediment which can reach up to several 10s % at the sea floor.

The reviewer indeed makes a very good point here, and comparison of the sediment volume/mass eroded in the Congo Canyon from 2019-20 and global riverine sediment fluxes did need further

tightening up. Our thanks for highlighting this key point. First, the conversion of (wet) eroded volume to (dry) sediment mass does need to include the porosity of that eroded material volume, which is indeed filled with seawater. We now assume an average porosity of 60-80% volume in the upper ~20 m of sediment that was eroded, based on a global compilation of Kominz et al. (2011). Assuming a pore water density of 1,035 g/cm³ that gives a reasonable range of wet densities (1.1-1.6 g/cm³) that is broadly consistent with wet sediment densities reported in short cores from a few meters below the seabed on the fan (e.g. Croguennec et al., 2017) , and with global averages for seabed sediment wet density (Tenzer and Gladkikh, 2014). We also now use a dry sediment density of ~2.5 g/cm³, acknowledging that not all that sediment is quartz (e.g. 2-3% is OC), but we could not find more detailed density data there. This is all now discussed in more detail in a new Supplementary Discussion section, and summarised in new footnotes to Table 1.

The reviewer is also correct that global estimates of riverine sediment flux (e.g. Milliman and Farnsworth, 2011; or Syvitski et al. 2022) are based almost exclusively on suspended load measurements, as bedload is notoriously tricky to measure, such that there are (very) few bedload measurements. Bedload is often assumed to be '10% of the suspended sediment flux' but this is highly uncertain. Indeed, we summarise work from the mouth of the Congo River that suggests it could have a bedload flux of 130Mt/yr (Peter et al., 1978) which is far more than the suspended load estimate (43 Mt/yr) used in the papers by Milliman, Farnsworth and Syvitski et al. Thus, we now only compare eroded volumes with the global suspended sediment flux from all of the world's rivers. The suspended sediment flux in Milliman and Farnsworth (2011) is ~18,000 Mt, rather than ~19,000 Mt, albeit with additional bedload potentially on top of that 18,000 Mt value.

Since our initial submission, a major new review of global riverine sediment flux has been published by Syvitski et al. (2022). This new analysis suggests that human activities have halved the suspended sediment flux from rivers to the oceans (7,200 Mt/yr, as opposed to 18,000 Mt/yr in Milliman and Farnsworth). We thus use this most recent estimate of modern-day riverine suspended sediment flux in Table 1, and in our comparison to the mass of sediment eroded by these turbidity currents. However, we also acknowledge that all estimates of global riverine suspended sediment flux have a (very) large amount of uncertainty. Indeed, those uncertainties could well rival or exceed those from our estimate of eroded sediment mass in Congo Canyon (also see below). Thus, we also quote the higher older estimates from Milliman and Farnsworth (2011) for the (pre-Anthropocene) riverine suspended sediment flux in Table 1, and there is also a new Supplementary Discussion section. Note that this lower estimate of global riverine suspended sediment flux in Syvitski et al. (2022) largely

offsets a reduction in the estimated mass of sediment eroded in the canyon in 2019-20 (due to the incorporation of 60-80% porosity), so that the eroded mass is still a significant fraction of global flux.

Time-lapse bathymetry and the resulting difference maps can result in significant errors of estimated eroded volumes. Please indicate the error ranges that these are associated with and explain your error calculation in the Methods section.

We also greatly appreciate this pertinent comment, and we have now undertaken a much more thorough analysis of the uncertainty in bathymetric changes between surveys, and their effect on the final estimate of eroded volumes. We now adopt the same method used by Mountjoy et al. (2018) [reference 20] and advocated by Schimel et al. [new reference 70], using individual spatially variable uncertainty values for each grid cell generated by the CUBE-algorithm typically used to process multibeam data. Briefly, this method reports eroded volumes in the form of $X > Y$. X is a most probable value and uses changes in elevation from all grid cells, assuming that measurement errors are close to being symmetrically distributed about zero. Y is a minimum estimate of eroded volumes based on a 'limit of detection' below which elevation changes are not included in volume estimates. As for Mountjoy et al. (2018), we use a rather conservative ($k=1.96$) limit of detection based on the CUBE-derived spatially variable uncertainty for each grid cell.

The new Methods section now includes:

Volume and mass of seabed sediment eroded along the Congo Canyon-Channel in 2019-20

Patterns and volumes of seabed erosion along the Congo Canyon-channel were determined using the 2019 and 2020 swath multibeam surveys (Fig. 1). Changes in elevation were multiplied by grid cell areas to derive volume. Volumes of seabed change did not include the lobe, beyond the end of the deep-sea channel.

Four methods were trialled to determine volumes of seabed change (Supplementary Figs. 9 and 10). The first three methods define a 'limit of detection' for real seabed change, and values below this limit are then discarded when calculating eroded or deposited seabed volumes. Importantly, these three methods produce volumes of seabed change that are minimum values. The limit of detection may either be spatially uniform (at least within each of the upper-canyon or lower-channel survey areas), or spatially varying with a unique value being assigned to each grid cell⁷⁰. A fourth method assumes that measurement errors are symmetrically distributed about zero, and these errors will thus cancel out over the survey areas. This final method thus returns a 'best guess' for volume of seabed change, rather than a minimum value.

Method 1: Changes in seabed elevation were measured for areas that are assumed to have undergone no significant ($\ll 1$ m) change from 2019-20. These areas were located outside the main canyon-channel axis (Supplementary Figs 7 and 8). Histograms of seabed changes in these areas (Supplementary Figs 7 and 8) are then used to define a 'limit of detection' for real seabed change in other parts of the same survey. This analysis suggests that changes in seabed elevation between surveys of < 4 - 6 m in the upper canyon, and < 10 - 15 m in the lower channel, can be caused by measurement errors (Supplementary Figs 7 and 8).

Method 2: It is often assumed that uncertainties in seabed elevation for individual surveys are $< 0.2\%$ to $< 0.5\%$ of the water depth⁷⁰, which would lead to uncertainties of 3 to 10 m in the upper canyon (~ 1.5 to 2 km water depth), and 6 to 25 m in the deeper-water (~ 3 to 5 km) channel (Fig. 1). Uncertainties from both surveys then need to be combined when calculating the limit of detection in seabed change between surveys⁷⁰. Thus, assuming uncertainties from each survey are summed, this method gives a limit of detection for seabed elevation change of 6-20 m in the upper canyon, and 12 to 50m in the lower channel. Method 1 (Supplementary Figs 7 and 8) suggests that the lower range of these estimates are most likely, for these surveys with narrow beams. Non-random spatial patterns of seabed change $> \sim 5$ m in the upper canyon (Supplementary Fig. 7d,e), and > 15 m in the lower channel (Supplementary Fig. 8d,e), which are physically reasonable (e.g. non-random and focussed along the canyon floor only), also suggest a reasonable limit of detection is closer to the lower end of this range estimated by method 2.

Method 3: The CUBE algorithm implemented within software (CARIS) typically used to process multibeam echosounder data automatically provides an estimate of spatially varying uncertainties for different grid-cells⁷⁰. These CUBE-derived uncertainties include additional important factors, such as whether data come from inner or outer acoustic beams in the multibeam sonar array, and are thus preferable to other methods⁷⁰. CUBE derived uncertainty values are then combined in quadrature (to ensure all resulting values are positive) to derive combined uncertainties in elevation changes between two surveys⁷⁰. CUBE-derived uncertainties in changes in seabed elevation are typically less than 5 m in the upper canyon (Supplementary Fig. 9b,c) or 10-15 m in the lower channel (Supplementary Fig. 10b,c). The upper limits of the spatially variable CUBE-derived uncertainties are thus also broadly comparable to those derived via to Method 1. The CUBE-derived uncertainty value for each grid cell in the difference map can then be multiplied by a constant termed k ⁷⁰. A value of $k = 1$ ensures that raw uncertainties values calculated by CUBE are used as the limit for detection, whilst higher values of k are more conservative. Mountjoy et al.²⁰ used a value of $k = 1.96$ (two standard deviations or 95% confidence limits) to define a limit for detection⁷⁰. Higher values of k generate progressively more conservative limits of detection, and provide greater confidence that seabed

change is real. However, they also cause data from more grid cells to be discarded. Indeed, a sufficiently high value of k will conclude with ~100% percent confidence that at least zero seabed change occurred; which is not a useful conclusion⁷⁰.

Method 4: A final method assumes that measurement errors are symmetrically distributed about a zero value, and these errors will thus cancel out over the survey areas. This final method thus returns a 'best guess' for volume of seabed change, rather than a minimum value.

Error bars for volumes of seabed change: Methods 1-3 are based on limit(s) of detection below which measured values of seabed change are discarded when calculating volumes of change. These limits of detection can produce error bars for volume of seabed change, via multiplying the area of the grid cell and the corresponding limit of detection. However, this approach can produce very large ranges for error bars, which may indeed exceed the main estimate of seabed volume change⁷⁰. Importantly, such an approach implicitly assumes that errors can reach maximum values simultaneously at every grid cell, and that errors are thus not close to being symmetrically distributed about zero. Thus, and as noted by Schimel et al.⁷⁰, the significance of error bars derived by multiplying limit(s) of detection by grid cell area is thus uncertain.

Chosen method: In the main part of this paper we chose to follow the method of Mountjoy et al.²⁰ for reporting volumes of seabed change, and take into account some recommendations of Schimel et al.⁷⁰ We report volume changes in the format $X [>Y]$, where X is a 'best guess' that is simply based on changes in seabed elevation measured at all grid cells (i.e. Method 4). This 'best estimate' assumes that measurement errors are close to being symmetrically about zero, and thus cancel out. However, we also then report a minimum estimate for volume of seabed change (Y), which is based on a limit of detection. We choose to use the CUBE-derived uncertainties in seabed elevation change at each grid cell, as they incorporate a wider range of important uncertainties than other methods⁷⁰. We also chose a value of $k = 1.96$ (Method 3) following Mountjoy et al.²⁰, and note that this is a rather conservative limit of detection⁷⁰. In supplementary figures 9 and 10 we therefore also show the volumes of seabed change derived using all four methods, comprising (i) no limit of detection and using all values of seabed change in every grid cell, spatially variable limits of detection based on CUBE-derived uncertainties and a value of (ii) $k = 1$ and (iii) $k = 1.96$, and (iv) a spatially uniform limit of detection that is 5 m in the upper canyon and 15 m in the deep-water channel. The volume of seabed change is also split into volumes of erosion and deposition (Supplementary Figs 9 and 10), although deposited volumes are often much smaller. This shows how the different methods affect

(sometimes minimum) volumes calculate for erosion, deposition and net change (Supplementary Table 4).

We explain how volumes are reported to the reader (X [$>$ Y] as above) where volumes are first cited in the main body of the text. There was no space to add that text to the abstract, but we think this is still ok, as there is an explanation in that main body, and a full description in the methods section.

In the original manuscript, we also stated that the time-lapse swath surveys in 2019-20 of the upper canyon and international waters covered 50% of the canyon-channel's total length. However, this was incorrect, as they actually covered 40% of that total length. Given that erosion is often deep ($>$ 20 m) in both the proximal and distal time lapse surveys, it is reasonable to assume that comparable levels of erosion occurred in the intervening part of the canyon that experience flushing flows with broadly similar front speeds. Thus, we felt it was appropriate to state both the eroded volume/mass surveyed (i.e. along 40% of the system), and then extrapolate that to an estimate of the eroded volume along the entire (100%) of the system length. We feel that this assumption is warranted, and thus now also forms part of the total range for the eroded mass, which is expressed as $>1.07 \text{ km}^3$ [$>$ 0.40 km^3] (surveyed reaches) to $>2.68 \text{ km}^3$ [$>$ 1.00 km^3] (extrapolated to the entire system length).

As you are claiming that these very large sediment volumes are redistributed in just a single year, the errors on that estimate (and therefore possible minimum and maximum volumes) are important. To make this a fair comparison, these factors should be mentioned and corrected for as much as possible. For now, I think the eroded volume by turbidity currents making up "14-28% of the annual global (suspended?!) sediment flux from rivers" is a maximum estimate that may need some down tuning. However, eroded volumes by turbidity currents in the Congo are very large!

In general, we agree that the main big picture point being made is simply that the eroded volume and mass is large, and it would indeed be possible to lose sight of that being the main point.

However, as suggested, we have now undertaken a much more thorough analysis of the uncertainty in bathymetric changes between surveys, and their effect on the final estimate of eroded volumes. We adopted the same method used by Mountjoy et al. (2018) [reference 20] and advocated by Schimel et al. [new reference 70], which reports eroded volumes in the form of X [$>$ Y]. X is a most probable value and uses changes in elevation from all grid cells, assuming that measurement errors are close to being symmetrically distributed about zero. Y is a minimum estimate of eroded volumes based on a 'limit of detection' below which elevation changes are not included in volume estimates.

As for Mountjoy et al. (2018), we use a rather conservative ($k=1.96$) limit of detection based on the CUBE-derived spatially variable uncertainty for each grid cell. We also revised the method for converting from (wet) sediment volume to (dry) mass fluxes, as also set out above. We felt it was appropriate to use the most recent estimate of present-day (Anthropocene) sediment river flux in Syvitski et al. (2022), although Table 1 also includes estimates from older work. We definitely agree that estimates of riverine bedload flux are extremely uncertain, thus we restrict comparisons in the main text to the suspended sediment flux of $\sim 7,200$ Mt/yr of Syvitski et al. 2022. At the end of this new analysis we conclude that the mass of sediment eroded in 2019-20 is 1,338-2,675 Mt [> 535 -1,070 Mt], which makes it 19-37% [>7 -15%] of the present-day global suspended sediment flux from rivers given by Syvitski et al. (2022; i.e. 7,200 Mt/yr). Our thanks for this comment that caused us to thoroughly revise our method, and at the end of that process we then derived 19-37% [>7 -15%].

2. Representation of frontal speed measurements

And in order to assess the speed of ignition and a possible steady state velocity, it is important to report the error bars and variability in these measurements. Please add these in the main text, methods and figures. Frontal speeds of the turbidity currents are estimated by the arrival times at the mooring & distance in between moorings.

This is another very good point, and it was indeed important to include a better analysis of the uncertainties in the flow front speeds. Our apologies there. These uncertain are now outlined in Methods section, and included on Figure 3 (where they exceed the thickness of the line on the plot). The ADCPs recorded velocity profiles every 9-12s (moorings M1-M8) or 45s (mooring A2 and A3), whilst clock drift for the ADCPs was negligible (<225 s in 6 months; $< 0.0014\%$ of times). The timing of cable breaks is known to the nearest minute. The main uncertainty in front speed is thus typically due to $<2\%$ uncertainty in distances measured between moorings, which arises from operator choice as to exact location of the deepest part of the canyon-channel floor. We calculated this to be a 2% uncertainty by having different people re-measure the distance between the same mooring sites. The only exceptions are front speeds from moorings M2 to M9, located just ~ 370 m apart, where the frequency of ADCP measurements becomes important, leading to larger percentage error bars. It is also now noted that front speeds from cable breaks assume that the cable is immediately broken by the arrival of the flow's front, whilst those from moorings assume its position was not changed by previous flows. These assumption are now noted in the Methods section.

But how do these velocities compare to the actual measured ADCP velocity profiles? Do they agree?

A detailed analysis of ADCP data is being written up as a separate paper, led by an ECR in the wider team, and we do not want to duplicate their work; so ADCP data on internal speeds is not included here. However, it is also important to note that ADCPs only recorded internal speeds for a few flows, which are all in the upper canyon, due to the way in which flows broke mooring anchors (fig. 2). The comparison can thus only be made for a few of the slower moving flows in the upper canyon. An initial analysis shows internal (ADCP) speeds are consistent with front speeds outlined here for those upper canyon events (1-3 m/s), but this will be part of the subsequent paper led by the ECR.

3. Threshold initial front speeds & grain size

The authors pose the new working hypothesis that threshold initial front speeds (for ignition) are not strongly depended on grain sizes. This is a very interesting finding that needs further testing.

However, it indicates that these huge muddy and carbon rich systems (e.g., the Congo system) can be equally efficient in transporting material to the deep sea as sand-dominated systems. In order to strengthen this point, it may be useful to incorporate actual grain sizes measurement into e.g., fig 6.

This was a very useful comment about an important conclusion, which we considered carefully. We did not include specific grain size values on Fig. 6 (new Fig. 7), but we did provide estimates of the sand and mud percentages for each system on new Fig. 7 (and as requested by Reviewer 2). Indeed, we added a range of further information to new Fig. 7, which helps the reader compare the nature of these systems. We also added a detailed Supplementary Discussion on grain sizes in the systems compared, which notes the difficulties in assigning precise grain size ranges to each system. For example, field data on grain size distribution come from a mixture of sediment inputs (e.g. rivers), deposits (seabed cores) and traps in the flow (but whose height in the flow can vary due to tilts). Then, grain size distributions within a turbidity current will vary with height above the bed, and with distance from front to back of the flow, further complicating definition of 'representative' grain sizes. Thus, it can be unclear which parts of a grain size distribution, and from which part of a turbidity current, should be used to test past ignition theory. For these reasons, we only summarise approximate fraction of sand ($> 63\mu\text{m}$) and mud ($< 63\mu\text{m}$) present at the four field sites (Supplementary Table 3). It is then possible to determine which systems carry significantly more sand or mud than others, even though it is not possible to define exact grain size ranges carried within specific parts of turbidity currents. Other members of the project team are working on results from the (single) sediment trap thus far deployed in our project, and that will be part of later papers.

This comment then raises the implications of our results for the efficiency with which material is transported to the deep-sea by turbidity currents. This is a perceptive and important comment, which led us to make further changes. Grain size indeed seems to have little effect on initial threshold front speed needed for ignition-autosuspension. However, once ignition-autosuspension occurs, muddier flows (e.g. > 1,100 km in Congo Canyon) then travel much further than sandier flows (~50 km in Monterey Canyon), and eroded very different amount of sediment. So, once ignition-autosuspension has occurred, then grain size could then potentially affect efficiency and distance of material transfer to deep sea.

More generally, this comment made us realise that there are actually three important conclusions from the comparison of flow front speeds in four locations worldwide (old figure 6, new figure 7), not just one. Thus, we amended the text and added a new figure (new figure 8) to emphasise that as well as (1) the threshold initial front speed (4-5 m/s) for ignition-autosuspension is weakly affected by grain size, that (2) once flows have ignited-autosuspended, flow front speed is a rather poor predictor of final runout and degree of seabed erosion (i.e. scale of the flow, and thus efficiency of material transport). Flows with similar front speeds can travel for very different distances (e.g. 50 km v. 1,100 km). In addition (3) changes in flow speed are often rather gradual, despite in some cases eroding a very large volume of sediment, so fronts may approach an equilibrium-autosuspension state. The second point also has some important follow on implications, such that once ignited, similar flow fronts have very different bodies, and thus that flows front and body are in 'rather poor communication'. We did not have space in the main text to explore these points (we tried - they are rather wordy to explain), but they may well form a basis for subsequent more detailed publications.

4. Estuarine turbidity maximum

The exact nature and sediment-trapping process by an "estuarine turbidity maximum" remains somewhat unclear in the explanation and the supplementary figure 4d. Can you expand on the explanation and it's illustration in the figure here to clarify?

This is a reasonable request, and we have expanded the Supplementary Discussion material on how estuarine turbidity maximums tend to trap sediment, and how this could lead to turbidity currents at the mouth of the Congo River. Unfortunately, we did not have room to add more discussion of this process to the main text (we tried...), and asked readers to look at the supplementary discussion.

5. Use of colour scales

The (mis-) use of colour maps that visually distort data through uneven colour gradients or are not readable by people with colour-vision deficiency has been a matter of ongoing discussion (e.g., Crameri et al., 2020). The rainbow colour scale is one of these non-scientific colour scales. I recommend to get rid of the rainbow and use a scientific colour scale (e.g., from this library <https://www.fabiocrameri.ch/colourmaps/>) in all figures of the main manuscript and the supplement (see word documents for details.)

This is a very good suggestion, and we now employ the scientific colour scales provided by the library at <https://www.fabiocrameri.ch/colourmaps/>

Responses to Reviewer 2 (David Piper)

We greatly appreciated comments from the reviewer, which are as always exceptionally useful. We agree that these flows are far larger than those previously monitored in places such as Monterey Canyon and Canadian fjords, and this scale of flow will be much more important for building deposits in ancient turbidite sequences. We also acknowledge the efforts of various colleagues who helped to recover the drifting ADCP-moorings in 2020 (....somewhat against the odds).

The paper is irritating to read because of the seeming obsession of being the first to do something. Pointing it out once is useful, but it would be better to emphasize the scientific significance of the findings. The paper is a little repetitious between the front end and the back end.

As background, we had sent an initial draft paper out to several colleagues in Durham, who are not working on turbidity currents. One of those colleagues felt that the original draft 'read like an undergraduate literature review', and it was 'entirely unclear to them what was novel and thus important'. This rather strong feedback caused us to make it clearer what was novel, and we may have now overdone these statements of novelty. However, novelty may be clearer to those working on turbidity currents (such as the two reviewers), and be somewhat less clear to some of the wider audience for Nature Communications. Thus, we have toned down the statements of novelty, yet still briefly explain what is novel for that wider Nature Communications audience. In particular, we have:

- Removed the following statement from the start of discussion (old lines 200-023). 'Here we discuss the first detailed direct measurements from turbidity currents in the deep (> 2 km) sea. These unique measurements show how sediment can be transferred efficiently from a major river mouth to water depths of ~5 km, by the longest runout sediment flows (of any type) yet measured in action on Earth'.

- The following section has been modified from (old) lines 143 and 144 in the Study Area section. It is changed from 'However, no previous study had deployed ADCP-moorings at multiple sites extending from upper canyon to the deep sea, as occurred during this project in 2019-20209' to 'However, no previous study had deployed ADCP-moorings at multiple sites to the end of a deep-sea canyon-channel, as occurred during this project in 2019-2020'. This is because recent flow monitoring in Monterey Canyon also had multiple moorings, but only to 1.85km depth. That set of moorings did not extend to the deep (2-5 km) sea, nor to the end of a canyon-channel (the last Monterey mooring was within the upper canyon).
- We removed 'for the first time' from lines 205 and 206 in the Discussion. It now reads 'Here we document directly that major river floods generate powerful and long runout turbidity currents that flush very large amounts of sediment through submarine canyons'.
- We amended (old) lines 314-315 in the Discussion by adding the word 'some', so that it reads 'contrary to some previous models'.

It would be useful if the term "sediment" were qualified. What size sediment is delivered to the Congo mouth (x% fine sand, y% mud). What is the sediment type reworked from the floor of the canyon – sand or mud? What is the sediment type deposited on the lobe. Older turbidite literature often claims differences between sandy flows (Squamish, Monterey) and muddy flows (?Congo, Bengal) in flow characteristics and architecture. The brief statement on line 294 should have come earlier and should have been a bit more quantitative. It could be summarized in part from the Supplementary Material referencing the more detailed discussion there.

This is a useful suggestion, and echoes that of Reviewer 1. We have therefore included a detailed Supplementary Discussion section summarising grain sizes present within each of the four systems compared on fig 6 (new fig 7). This information includes the sand and mud fraction supplied to the system, that within sediment cores from canyon-channel axis or within traps, and that found in the lobe at the end of the Congo Canyon-channel by Dennielou et al. (2018). Indeed, we added a range of further information to new Fig. 7, which helps the reader compare the nature of these systems. Supplementary Table 3 also now summarises a range of differences between the various field sites.

Past work does indeed hypothesise that this sand/mud ratio can profoundly affect flow behaviour and efficiency of material transfer to the deep sea. As with Reviewer 1, this comment made us realise that there are actually three important conclusions from the comparison of flow front speeds in four locations worldwide, not just one. Thus, we amended the text and added a new figure (new figure 8) to emphasise that as well as (1) the threshold initial front speed (4-5 m/s) for ignition-

autosuspension is weakly affected by grain size, that (2) once flows have ignited-autosuspended, flow front speed is a rather poor predictor of final runout and degree of seabed erosion (i.e. scale of the flow, and thus efficiency of material transport). Flows with similar front speeds can travel for very different distances (e.g. 50 km v. 1,100 km). In addition (3) changes in flow speed are often rather gradual, despite in some cases eroding a very large volume of sediment, so fronts may approach an equilibrium-autosuspension state. The second point also has some important follow on implications, such that once ignited, similar flow fronts have very different bodies, and thus that flows front and body are in 'rather poor communication'. We did not have space in the main text to explore these points (we tried), but they may well form a basis for subsequent publications.

The authors seem to place great significance on discovering that large flushing events are not triggered by earthquakes. Perhaps the literature is heavily weighted towards earthquake triggering of turbidity currents, but it is not surprising that large canyon flushing flows have multiple origins. It is what has been shown by Bailey et al. in Monterey canyon, inferred from core data at Squamish. And was pretty clear in La Jolla canyon and fan in 1970. I recall a Gaoping Canyon study which had a tc several days after the flood peak.

We agree that it was perhaps not too surprising that large canyon flushing flows have multiple origins, and indeed felt that the literature has indeed tended to focus too much on earthquakes as their origin, especially as it is proposed that turbidites provide long term earthquake records in the deep-sea. Hopefully, this paper will itself start to address such a general imbalance.

It is also important to acknowledge that a range of previous work has proposed that river floods can generate turbidity currents that reach the deep sea, albeit using less direct evidence such as turbidite deposits and channel geomorphology. We thus added a new Supplementary Discussion section that summarises such work, and the main text references Piper and Normark (2009) and Thierry Mulder, as well as signposting readers to the longer supplementary section. We initially tried to include a more discussion in the main text, but there was just insufficient space. We therefore just stated in the introduction that *"However, it was not clear whether river floods also generate such large canyon-flushing events (Supplementary Discussion)"*, and started the discussion with *"However, it was uncertain whether river floods could also generated turbidity currents that flushed large amounts of sediment through canyons and into the deep sea (Supplementary Discussion); and if so, how this occurred"*. That previous work was indeed based on lines of evidence that are more uncertain than the direct monitoring data presented here, as the reviewer notes. For example, we agree with Piper and Normark (2009) that it is highly challenging to infer flood triggers from turbidite

deposits, as indeed supported by the recent monitoring work by Heerema et al. (2022) from offshore the Var River. We also felt that relatively straight submarine channels may also be formed by mechanisms other than flows from floods, such as migration of knickpoints (as shown in recent work from Bute Inlet and elsewhere). Our study also different from this past work in other key regards around how floods trigger turbidity currents. For example, past work suggested that floods triggered these turbidity currents via direct plunging of hyperpycnal river discharge, whilst we do not invoke such a hyperpycnal mechanism. So, as well as having much more direct and reliable field evidence of river floods triggering turbidity currents that reach the deep sea, there are different processes being invoked by this study at the river mouth.

We also note that the study by Bailey et al. (2021) of turbidity current timing in Monterey Canyon documented a canyon-filling turbidity current, rather than a canyon-flushing event, as these flows were restricted to upper part of Monterey Canyon. Work on La Jolla canyon has played a seminal role in understanding turbidity currents, in our view. Inman et al. (1972) note how ocean waves and internal tides may trigger flows, but these direct observations limited to shallow water in uppermost canyon. Piper (1970) used cored deposits, including in deeper water, but inferred a flood trigger based on deposit character, and this again seems to be based on more uncertain evidence (as above).

However, as the reviewer correctly points out, past work based on cable-breaks in Gaoping Canyon off Taiwan has shown directly that fast moving turbidity currents linked to river floods can run out for > 300 km into the deep sea (e.g. as in Fig. 7). Specifically, two sets of cable-breaks records turbidity currents triggered by floods associated with Typhoon Morakot in 2009 and Typhoon Soudelour in 2015. As the reviewer says, the Morakot-flood turbidity current was triggered 3 days after the flood, whilst the Soudelour-flood turbidity current appears to coincide with a (...much smaller) flood peak. There were no time lapse surveys available offshore Taiwan, so it was uncertain whether these flows flushed large amounts of sediment from the canyon, and the information was restricted to cable-breaks without confirmation of flow timing via ADCP-moorings. However, these events off Taiwan are indeed direct evidence of long runout turbidity currents after major floods, and they are now discussed in more detail in a new Supplementary Discussion section. For example, a key point is that the Morakot Flood is extremely large (carrying 280-570 Mt of sediment alone), whilst the Soudelour event shows how rather small floods (as for the newest Jan 2022 event in the Congo system) can produce turbidity currents at the same time as the flood peak.

It would have been useful to point out the analogy with Squamish in the spring tide influence on sediment instability (although the process may have been different, e.g. pore pressure as in the Fraser delta).

We absolutely agree that a body of recent work at a set of river-deltas in Canada provides relevant insights into how a combination of elevated river discharge and spring tides may initiate turbidity currents. These studies document how elevated river discharge and low (ebb) tides produce much stronger offshore-direct river plumes, that carry greater amounts of suspended sediment, which then settles onto the seabed. As the reviewer also states, there are also some key differences. The turbidity currents in these Canadian studies tend to coincide with elevated river discharges, whilst some Congo Canyon turbidity currents occurred much later at rather low river discharge (e.g. April 2021). This comparison is now a Supplementary Discussion section on “Recent monitoring showing how turbidity currents may be caused by river floods and spring tides”. We also changed the main text to say “Recent monitoring studies have shown how elevated river discharge and tides combine to generate much shorter runout (1-50 km) turbidity currents offshore from smaller river mouths, and how the threshold suspended sediment concentration of rivers needed for offshore flows is much lower than once thought” (new lines 213-215).

In lines 354-357, there is some speculation on patchy erosion and knickpoints. This sounds like the evidence for “cyclic steps”, reflected in bedforms at Squamish. Should the authors comment on this possibility.

This pattern of patchy erosion will be the focus of a forthcoming paper led by one of the ECRs in the team, and we also do not have much space here to discuss this (interesting) question. However, the knickpoints seen in the Congo Canyon data are at a very different scale to the cyclic step bedforms (wavelengths of ~60 m, amplitude of 2-3 m) reported by Hughes Clarke, Hage and others at Squamish Delta, and unambiguously linked to cyclic steps in supercritical flow. Indeed, due to the deeper waters of this Congo Canyon study area, a ship-based multibeam survey would not resolve such smaller-scale (2-3m high) features, and we hope to obtain AUV surveys of the Congo Canyon floor in 2023 with colleagues at IFREMER. The patchy erosional features in the Congo Canyon more closely resemble the knickpoints described by Heijnen et al. (2020) in Bute Inlet, although some of the patches of erosion in Congo Canyon seem to lack a distinct headscarp at their upstream end. It is less clear whether those knickpoints are linked to cyclic steps, and this would need a more in depth discussion in the forthcoming paper.

I have no specific comments on the Supplementary Material, except to note that a little of what is

covered there in the text could usefully be moved to the main paper. The space for this could be achieved by a bit of tightening of repetition and claims of priority.

Our thanks for specific suggestions on where to cut material, to make way for the additional text in response to the previous points. We have indeed removed some text from the introduction that repeated later sections, and it was still challenging to reduce the main text to <5,000 words again.

We cut the following section from the introduction, as the information is given later in the paper.

“Two further long runout turbidity currents broke seabed telecommunication cables again on March 9th 2020 and 28-29th April 2021 (Supplementary Table 1), whilst the mooring array recorded twelve slower and shorter runout turbidity currents within the upper canyon during a ~4 month period in 2019-20 (Fig. 2)”. We shortened the introductory section on cable breaks, as the list of times for cable breaks was noted later. The introduction now states “Understanding why these cables broke repeatedly in 2020 and 2021 is crucial for assessing and potentially mitigating hazards due to submarine flows, especially as cables had not broken in the previous 18 years (Suppl. Table 1)”.

Responses to the various points noted on the original document are as follows (line numbers refer to the originally submitted document, not the revised version). We also had to shorten the text in places, such that it is now exactly at the 5,000 word limit for the main text.

- Line 37 (abstract): This is most likely consolidated material, especially given the depths (> 10-20m) of erosion, so we retain the term ‘eroded’.
- Line 41: hyphen removed.
- Line 67: ‘incinerated’ is changed to ‘oxidised’, which is indeed more accurate.
- Line 139: No change is made because the acronym ADCP was spelled out on line 96, when first used. A brief summary is also provided in lines 96-97 about what the instrument does.
- Line 155-156 repeats the statement that ‘cables had not broken in the last 18 years’, which was made previous on lines 83-84 in the introduction. We thus removed the statement in lines 83-84. However, this is a critical point that underpins a primary conclusion that unusually large floods in 2019/20 and 2020/21 are associated with the cable breaks. We thus amended figure 3, as we had run the paper past colleagues, some of whom has missed this critical point about no breaks in previous 18 years etc.
- Line 164-165 states ‘two weeks after the flood peak when river discharge was high’. We agree that this is unclear, and it is now changed to ‘The first cable-breaking flow occurred on January 14-16th, three weeks after the flood peak in December 21st, albeit when river

discharge was still relatively high (Fig. 3b). The arrival times of this January 14-16th turbidity current were captured by eight ADCP-moorings just before they broke’.

- Line 171: This is a good point that 4.5 months is far from immediate. We therefore amended the text to say ‘which were triggered finally at spring tides that occurred 3 weeks to 4.5 months after the flood peaks (Fig. 4)’.
- Line 193: The reviewer asks a good question that ‘Do you know if the material eroded was unconsolidated sediment filling the canyon, or was it from the "bedrock" (Pliocene - Quaternary) canyon floor’. This is difficult to answer with certainty as the depth of erosion (often > 20m) is much deeper than the length of available dated cores in the upper canyon, and we thus make no changes to the text here. Previous work by Picot et al. (2016, 2019) indicates that channels in deep-water are <200 ka in age, but their study did not include the upper canyon. Our expectation is indeed that eroded sediment in both the deep-water channel and upper canyon is much younger than Pliocene-Quaternary bedrock, and this may be something to discuss further in later papers perhaps.
- Line 207: We prefer to retain the term ‘eroded’ rather than ‘reworked here’, because this is the amount of sediment missing (i.e. eroded) between surveys, and we thought eroded was thus more easily understood. But we agree that this eroded sediment was also then reworked into deeper-water, so either term might be used.
- Line 220: This is a good point, and the reference to ‘first’ here is removed.
- Line 289: The reviewer makes a good point, for an event they know particularly well. “If this flows evolved from small debris flows then there must have been an initial phase of acceleration (not captured by cable breaks) prior to the first cable break. So 1929 (and Nice tc) may not be different, they just lack data in the canyon flow prior to flow expansion.”. We thus rephrased the section to say - as do sometimes (rather than initially) faster flows (e.g. NW Atlantic turbidity current).
- Line 311. The reviewer asks - ‘Is there really such an emphasis on earthquakes in the literature’. The lead authors impression is that there is a widely held view that earthquakes play a key role in triggering larger and faster turbidity currents, and this view underpins a number of papers on using turbidites as earthquake-records (e.g. see work ranging from Goldfinger et al., 2003 to Howarth et al., 2021). As outlined above, we tried to address this balance by adding more references to past work (including by this reviewer and Bill Normark and Thierry Mulder) on how river floods may generate larger and faster flows that reach the deep-sea, and via a new Supplementary Discussion section. In the main text we had little space, but state that “However, it was not clear whether river floods also generate such

large canyon-flushing events (Supplementary Discussion)”, and started the discussion with “However, it was uncertain whether river floods could also generated turbidity currents that flushed large amounts of sediment through canyons and into the deep sea (Supplementary Discussion); and if so, how this occurred”.

- Line 334: In the full data set, erosion tends to occur along the canyon channel-floor, but we agree that there is also significant contributions from related side-wall failures - as shown in old Figure 7 (new Figure 6). Thus, we have rephrased this sentence to say: “It also emphasises how turbidity currents physically disturb benthic fauna, as tens of meters of sediment may be removed locally along the canyon-channel floor, sometimes with related side-wall failures (Fig. 7). ”.
- Table 1: Thanks for spotting this error, and it is corrected to N.W. Atlantic.

Our thanks again for a set of challenging and perceptive reviews, and we hope that these responses have now strengthened the paper significantly. Your time and efforts were really appreciated.....

Best wishes

Peter Talling (on behalf of the authors).

REVIEWERS' COMMENTS

Reviewer #1 (Remarks to the Author):

Remarks to the Authors

I highly appreciate the detailed answers of the authors to my comments and the large amount of work and diligence that the authors have put into the manuscript during this review process.

This time was well spent and the manuscripts' clarity has improved. Especially the models for turbidity generation are now clear and well-illustrated.

I am happy with the manuscript in its current form and it can be published almost as is.

Only 3 tiny comments that occurred while re-reading:

1. Paragraph starting at Line 298:

Initial front speeds as a poor predictor for eroded sediment volume: As the two canyon systems compared – the Congo and the Monterrey system – are characterized by very different grain size distributions along the canyon and channel floor (see paragraph before in the revised manuscript), the eroded volume will also depend on these initial seabed conditions. Loosely packed clay-rich sediment may be easier to erode than coarse sand and/or tightly packed cohesion-dominated clay-rich material may be more resistant to erosion than loose sand.

2. Line 321 insert 'of' between 'clusters canyon-flushing'

3. Line 453-455 and line 490-392 are repetitive.

Thank you for giving me the opportunity to review this excellent manuscript.

All the best,

Anne Bernhardt